# RIGID INVARIANT SLICED WASSERSTEIN VIA INDEPENDENT EMBEDDINGS

**Zakk Heile**[*]    **Peilin He**[*]    **Jayson Tran**[*]    **Alice Wang**[*]    **Shrikant Chand**

[1]Division of Natural and Applied Sciences, Duke Kunshan University
[2]Department of Computer Science, Duke University
[3]Department of Mathematics, Duke University

zakk.heile@duke.edu
peilin.he@dukekunshan.edu.cn
jayson.tran@duke.edu
alice.wang@duke.edu
shrikant.chand@duke.edu

[*]Equal contribution

## ABSTRACT

Comparing probability measures modulo unknown rigid transformations is a central challenge in geometric data analysis. Classical optimal transport (OT) distances, including Wasserstein and sliced Wasserstein, are sensitive to rotations and reflections, whereas Gromov-Wasserstein (GW) and Procrustes-Wasserstein (PW) distances are invariant to isometries but computationally prohibitive for large datasets. We introduce *Rigid-Invariant Sliced Wasserstein via Independent Embeddings* (RISWIE), a scalable distance that combines the invariance of NP-hard approaches with the efficiency of projection-based OT. RISWIE utilizes data-adaptive bases and matches optimal signed permutations along axes according to distributional similarity to achieve rigid invariance with nearly linear complexity in the sample size. We prove bounds relating RISWIE to GW in special cases and demonstrate dimension-independent statistical stability. Our experiments on cellular imaging and 3D human meshes demonstrate that RISWIE outperforms GW and PW in clustering tasks and discriminative capability while significantly reducing runtime.

## 1 INTRODUCTION

Optimal transport (OT) distances, such as the Wasserstein distance, have become central tools in data analysis due to their ability to compare probability measures in a geometrically meaningful way (Peyré & Cuturi, 2019; Santambrogio, 2015). However, in settings where shapes may be oriented arbitrarily, the usefulness of these distances is limited, as they are sensitive to rigid transformations (Besl & McKay, 1992). Although rigid transformations preserve pairwise distances, rigid point set registration with unknown correspondences is computationally intractable: the problem requires jointly optimizing over rotations and permutations, yielding an NP-hard combinatorial search over point correspondences (Cela, 2013; Ling, 2024).

Several approaches address rigid misalignment by explicitly aligning point sets during comparison. Procrustes–Wasserstein methods jointly optimize over rigid transformations and transport plans, equivalently minimizing the Wasserstein distance over all relative orientations between two shapes (Adamo et al., 2025). In the absence of known correspondences, however, this formulation is NP-hard (Ling, 2024). More broadly, prior approaches to isometry-invariant shape matching optimize continuous relaxations of Gromov–Hausdorff–type distortions or rotation-invariant optimal transport objectives; these formulations are nonconvex and are solved in practice via iterative alternating optimization (Lai & Zhao, 2017; Bronstein et al., 2006).

Other approaches avoid explicit registration altogether by enforcing invariance to rigid motions by construction. The Gromov–Wasserstein (GW) distance (Mémoli, 2011) compares metric measure spaces via their internal relational structure rather than ambient coordinates. While theoretically

appealing, GW distances require solving high-order, nonconvex optimization problems and are NP-hard in general. Recent work has proposed entropic regularization and algorithmic accelerations to improve practical performance (Rioux et al., 2024; Zhang et al., 2024; Li et al., 2023), but these methods remain computationally demanding at scale and typically rely on approximate solvers. Consequently, the resulting distances are not guaranteed to preserve the ordering induced by the exact objective, potentially leading to inconsistent rankings across datasets.

Motivated by this trade-off between invariance and computational tractability, we introduce a new distance that achieves rigid invariance while remaining computable in polynomial time and scalable to large datasets.

**Contributions.** Our main contributions are:

(i) We introduce **RISWIE**, a sliced transport distance that combines data-dependent embeddings with optimal signed-permutation alignment to compare measures up to rigid transformations at nearly linear cost in the size of the empirical measures.

(ii) We establish theoretical guarantees, including rigid invariance, the pseudometric property, closed-form expressions for Gaussian measures, and explicit bounds relating RISWIE to Gromov–Wasserstein.

(iii) We demonstrate empirical, dimension-independent finite-sample convergence for bias and variance.

(iv) We show that RISWIE achieves state-of-the-art runtime with essentially no accuracy trade-offs in shape partitioning, clustering, and alignment benchmarks.

## 2 PRELIMINARIES

We use $\| \cdot \|$ to denote the $\ell_2$ norm on $\mathbb{R}^d$, $\mathcal{P}(\mathbb{R}^d)$ the set of Borel probability measures on $\mathbb{R}^d$, and $\mathcal{P}_2(\mathbb{R}^d)$ the subset with finite second moments. Given $\mu, \nu \in \mathcal{P}_2(\mathbb{R}^d)$, the 2-Wasserstein distance is

$$W_2^2(\mu, \nu) = \inf_{\pi \in \Pi(\mu,\nu)} \int_{\mathbb{R}^d \times \mathbb{R}^d} \|x - y\|^2 \, d\pi(x, y), \tag{1}$$

where $\Pi(\mu, \nu)$ is the set of couplings with marginals $\mu, \nu$ (Villani, 2008; Santambrogio, 2015). In practice, the above measures are approximated by the empirical sample-based measures

$$\mu_s = \tfrac{1}{s} \sum_{i=1}^{s} \delta_{x_i}, \quad \nu_t = \tfrac{1}{t} \sum_{j=1}^{t} \delta_{y_j},$$

which can be shown to converge weakly as $s, t \to \infty$ by a theorem of Varadarajan (Varadarajan, 1958). For $n$ samples, the computation of this distance scales as $O(n^3 \log n)$, and entropic regularization reduces this to $O(n^2)$ per iteration using Sinkhorn updates (Peyré & Cuturi, 2019). Despite these improvements, Wasserstein remains expensive in high dimensions and sensitive to rigid transformations.

An extension of this distance is the Procrustes–Wasserstein distance, which seeks a rigid transformation of the ambient space that best aligns two measures jointly with transport. Formally, it is defined as

$$\mathrm{PW}_2^2(\mu, \nu) = \inf_{R \in O(d)} \inf_{\pi \in \Pi(\mu,\nu)} \int_{\mathbb{R}^d \times \mathbb{R}^d} \|x - Ry\|^2 \, d\pi(x, y). \tag{2}$$

where $O(d)$ is the set of all rigid transformations. While this formulation is desirable, it is NP-hard (Ling, 2024).

Taking a different approach to achieve rigid-invariance, the Gromov–Wasserstein (GW) distance compares measures on metric spaces $(X, d_X)$ and $(Y, d_Y)$ without requiring a shared ambient space, instead aligning their internal distance structures (Mémoli, 2011):

$$\mathrm{GW}_2^2(\mu, \nu) = \inf_{\pi \in \Pi(\mu,\nu)} \iint \left| d_X(x, x') - d_Y(y, y') \right|^2 d\pi(x, y) \, d\pi(x', y').$$

While GW is invariant to rigid transformations, it also requires solving an NP-hard quadratic assignment problem (Cela, 2013; Kravtsova, 2025). Even approximate solvers scale as $O(n^3)$ per iteration, making GW computations scale poorly with sample size (Kerdoncuff et al., 2021; Rioux et al., 2024; Li et al., 2023).

While Procrustes-Wasserstein and Gromov-Wasserstein are the two natural rigid-invariant distances, our distance relies on nice properties of Wasserstein in 1D.

In one dimension, if $F_\mu$ and $F_\nu$ denote the cumulative distribution functions (CDFs) of $\mu$ and $\nu$, the $W_2$ distance admits the closed form

$$W_2^2(\mu,\nu) = \int_0^1 \left( F_\mu^{-1}(t) - F_\nu^{-1}(t) \right)^2 dt,$$

which can be evaluated in $O(n \log n)$ (Villani, 2008). The sliced Wasserstein (SW) distance extends this to higher dimensions by projecting onto directions $\theta \in S^{d-1}$ and averaging:

$$\mathrm{SW}_2^2(\mu,\nu) = \int_{S^{d-1}} W_2^2(P_\theta \# \mu, P_\theta \# \nu)\, d\theta,$$

where $P_\theta(x) = \langle x, \theta \rangle$ and $P_\theta \# \mu$ denotes the pushforward of $\mu$ under $P_\theta$ (Rabin et al., 2012; Kolouri et al., 2019).

## 3 METHODOLOGY

We now define a new distance, which we denote as the Rigid-Invariant Sliced Wasserstein via Independent Embeddings (RISWIE) distance, which we will show to minimize the trade-off between computability and desirability. The construction has three components: (i) data-dependent embeddings that map each distribution into a low-dimensional coordinate system derived from its own geometry, (ii) an alignment step that pairs axes across embeddings using signed permutations, and (iii) an aggregation of one-dimensional Wasserstein costs over the matched axes.

### 3.1 PROBLEM FORMULATION

Let $\mu, \nu \in \mathcal{P}_2(\mathbb{R}^d)$ be probability measures. We first define the object over which we optimize to define our distance, which is related to the concept of a rigid transformation.

**Definition 1** (Signed Permutation Group). The *signed permutation group* on $k$ elements is

$$\mathcal{O}_k^\pm := \{R \in \mathbb{R}^{k \times k} : R^\top R = I_k,\ R_{ij} \in \{0, \pm 1\},\ \text{one nonzero per row/column}\}. \quad (|\mathcal{O}_k^\pm| = 2^k\, k!)$$

Equivalently,

$$\mathcal{O}_k^\pm = \{D_\varepsilon P_\pi : \pi \in S_k,\ D_\varepsilon = \mathrm{diag}(\varepsilon_1, \ldots, \varepsilon_k),\ \varepsilon_j \in \{\pm 1\}\},$$

where $P_\pi$ is the permutation matrix associated with $\pi$, i.e., $(P_\pi)_{ij} = 1$ if $i = \pi(j)$ and $0$ otherwise.

The RISWIE distance defined below can be seen as the minimum cost axis and relative sign pairing across all $2^k k!$ pairings, where the cost is defined as the Wasserstein distance between the distributions embedded on those axes. We will denote it as $D$ throughout the paper.

**Definition 2** (RISWIE Distance). Let $\mu, \nu$ be centered probability measures on $\mathbb{R}^{d_1}$ and $\mathbb{R}^{d_2}$, respectively. Let $\phi := (\phi_1, \ldots, \phi_k) : \mathbb{R}^{d_1} \to \mathbb{R}^k$ and $\psi := (\psi_1, \ldots, \psi_k) : \mathbb{R}^{d_2} \to \mathbb{R}^k$ be fixed embedding functions. Let $\mathcal{O}_k^\pm$ denote the group of signed permutation matrices of size $k \times k$. For $R \in \mathcal{O}_k^\pm$, define $(R\psi)_j := \varepsilon_j \psi_{\pi(j)}$, where $R$ corresponds to a signed permutation $(\pi, \varepsilon)$.

The Rigid-Invariant Sliced Wasserstein via Independent Embeddings (RISWIE) distance is defined as

$$D^2(\mu,\nu) := \min_{R \in \mathcal{O}_k^\pm} \frac{1}{k} \sum_{j=1}^k W_2^2 \left( (\phi_j)_\# \mu,\ ((R\psi)_j)_\# \nu \right),$$

where $W_2$ denotes the 2-Wasserstein distance on $\mathbb{R}$ and $(\phi_j)_\# \mu$ is the pushforward of $\mu$ under $\phi_j$.

This definition only requires considering the relative sign difference between any two axes that are compared because $W_2$ is invariant under simultaneous reflection in one dimension. Thus, the minimization is equivalent to evaluating all possible axis pairings together with all possible sign assignments for each pairing. We additionally assume the distributions are centered with mean 0.

The embeddings $\phi_j$ and $\psi_j$ are user-friendly and may be obtained via linear (e.g. PCA) or nonlinear (e.g. diffusion maps) dimensionality reduction techniques (Coifman & Lafon, 2006), or other data-dependent procedures. This formulation avoids requiring a common projection basis, since alignment is performed directly between the one-dimensional pushforwards of $\mu$ and $\nu$.

The group $\mathcal{O}_k^\pm$ captures the necessary permutations and sign changes of embedding coordinates, corresponding to orthogonal transformations that preserve the independence of axes. Furthermore, minimization over $\mathcal{O}_k^\pm$ is a finite assignment problem solvable in $O(k^3)$ via the Hungarian algorithm, assuming pairwise costs have already been computed (Munkres, 1957).

---

**Algorithm 1:** RISWIE Empirical Computation

---

**Input:** Empirical measures $X = \{x_1, \ldots, x_{n_1}\} \subset \mathbb{R}^{d_1}$, $Y = \{y_1, \ldots, y_{n_2}\} \subset \mathbb{R}^{d_2}$;
      embeddings $\Phi = (\phi_1, \ldots, \phi_k)$, $\Psi = (\psi_1, \ldots, \psi_k)$.
**Output:** $D(X, Y) = D(X, Y)$.

$X \leftarrow \{x_i - \mathrm{mean}(X)\}_{i=1}^{n_1}$;    $Y \leftarrow \{y_i - \mathrm{mean}(Y)\}_{i=1}^{n_2}$

**for** $\ell = 1, \ldots, k$ **do**
    $A_\ell \leftarrow \big(\phi_\ell(x_1), \ldots, \phi_\ell(x_{n_1})\big)$;            `// embed X onto axis ℓ`
    $B_\ell \leftarrow \big(\psi_\ell(y_1), \ldots, \psi_\ell(y_{n_2})\big)$;            `// embed Y onto axis ℓ`
    $\widetilde{A}_\ell \leftarrow \mathrm{sort}(A_\ell)$;    $\widetilde{B}_\ell \leftarrow \mathrm{sort}(B_\ell)$;    `// sort in ascending order before`

**for** $\ell = 1, \ldots, k$ **do**
    **for** $m = 1, \ldots, k$ **do**
        $c_{\ell m}^+ \leftarrow \mathsf{W2sorted}^2(\widetilde{A}_\ell, \widetilde{B}_m)$;
        $c_{\ell m}^- \leftarrow \mathsf{W2sorted}^2(\widetilde{A}_\ell, \mathrm{reverse}(-\widetilde{B}_m))$;      `// reflect and reverse`
        $C_{\ell m} \leftarrow \min\{c_{\ell m}^+, c_{\ell m}^-\}$;        `// best sign for pair (ℓ,m)`

$\pi^\star \leftarrow \arg\min_{\pi \in S_k} \sum_{\ell=1}^k C_{\ell, \pi(\ell)}$;            `// solved by Hungarian`
$Z \leftarrow \sum_{\ell=1}^k C_{\ell, \pi^\star(\ell)}$;
**return** $D(X, Y) \leftarrow \sqrt{Z/k}$;

**Note:** $\mathsf{W2sorted}^2$ assumes its two input vectors are already sorted (ascending). For equal weights, it returns $\frac{1}{N} \sum_{i=1}^N (u_i - v_i)^2$ when the two lists are length-$N$; for unequal lengths/weights, it runs the standard two-pointer monotone coupling in $O(n_1 + n_2)$ time. Pre-sorting each projected list once (above) avoids re-sorting inside every 1D OT call, saving a factor of $k$. Negating reflects the distribution across 0; reversing ensures the reflected list remains sorted in ascending order.

---

To analyze time complexity, we take $d := \max\{d_1, d_2\}$ and $n := \max\{n_1, n_2\}$. We also assume that $k \le d$ and $n \ge d$, as is common in practice.

**For PCA embeddings,**

$$O\big(\underbrace{nd^2}_{\text{covariances}} + \underbrace{kd^2}_{\text{top-}k\text{ eigens}} + \underbrace{knd}_{\text{projection}} + \underbrace{kn \log n}_{\text{sort once}} + \underbrace{k^2 n}_{k^2 \text{ sorted } W_2^2 \text{ calls}} + \underbrace{k^3}_{\text{Hungarian}}\big) = O\big(nd^2 + dn \log n\big).$$

**For Diffusion Map embeddings,**

$$O\big(\underbrace{n^2 d}_{\text{kernel build}} + \underbrace{kn^2}_{\text{top-}k\text{ eigens}} + \underbrace{kn \log n}_{\text{sort once}} + \underbrace{k^2 n}_{k^2 \text{ sorted } W_2^2 \text{ calls}} + \underbrace{k^3}_{\text{Hungarian}}\big) = O\big(n^2 d\big).$$

Both of the above embedding choices are computationally efficient when used with the proposed scheme, with PCA-RISWIE being nearly linear in the number of samples. With $n \ge d$, these approaches are faster than standard Optimal Transport, Gromov-Wasserstein, and equivalent asymptotically to Sliced Wasserstein with $d$ projection axes. However, because random sampling can

perform poorly in higher dimensions, one might instead choose a superlinear number of axes (such as $d \log d$), in which case RISWIE becomes asymptotically faster.

## 3.2 THEORETICAL PROPERTIES

We verify that RISWIE meets the criteria specified in the preceding sections. The first result establishes rigid invariance under mild conditions on the embedding procedure. We then show that RISWIE is a pseudometric on $\mathcal{P}_2(\mathbb{R}^d)$. Additionally, we give a closed-form expression for Gaussian measures with PCA embeddings, compare it to the Gromov–Wasserstein distance with explicit bounds.

**Theorem 1** (Rigid-Invariance). *Let $\mu, \nu \in \mathcal{P}_2(\mathbb{R}^d)$, and $T(x) = Rx + t$ an affine transformation for $R \in O(d)$, $t \in \mathbb{R}^d$. Suppose either:*

*(i) (PCA) All nonzero eigenvalues of the centered covariance of $\mu$ are unique (so $\mu$ has finite second moments); or*

*(ii) (Diffusion map) The embedding returns the same set of eigenvectors (up to sign) for a given matrix (i.e., deterministic eigensolver for fixed input).*

*Then*

$$D(\mu, \nu) = D(T_{\#}\mu, \nu).$$

*In particular, $D(\mu, T_{\#}\mu) = 0$.*

Like the empirical Sliced Wasserstein distance, Theorem 2 shows that RISWIE is a pseudometric.

**Theorem 2** (Pseudometric). *For any $X, Y, Z \in \mathcal{P}_2(\mathbb{R}^d)$ and for any embedding procedure, the RISWIE distance is a pseudometric.*

Deciding whether two point sets are identical up to a rigid transformation is computationally intractable in general, as it requires solving a combinatorial correspondence problem with factorial complexity in the worst case (Chaudhury et al., 2015; Ling, 2024). Consequently, one cannot reasonably expect a computable distance to satisfy identity of indiscernibles modulo rigid transformations, since doing so would amount to solving an NP-hard registration problem. Nevertheless, rigid equivalence can still be recovered in important special cases—such as Gaussian distributions—as a corollary of the next result.

**Theorem 3** (RISWIE Distance for Gaussians under PCA Embeddings). *Let $A \sim \mathcal{N}(\omega_A, \Sigma_A)$ and $B \sim \mathcal{N}(\omega_B, \Sigma_B)$ be Gaussian probability measures on $\mathbb{R}^d$ with finite second moments so that they admit eigendecompositions $\Sigma_A = U_A \Lambda_A U_A^\top$ and $\Sigma_B = U_B \Lambda_B U_B^\top$, where $\Lambda_A = \mathrm{diag}(\lambda_1^A, \dots, \lambda_d^A)$ and $\Lambda_B = \mathrm{diag}(\lambda_1^B, \dots, \lambda_d^B)$ with $\lambda_1^A > \cdots > \lambda_d^A \geq 0$ and $\lambda_1^B > \cdots > \lambda_d^B \geq 0$. Denote*

$$\mathbf{a} := \left(\sqrt{\lambda_1^A}, \dots, \sqrt{\lambda_d^A}\right), \quad \mathbf{b} := \left(\sqrt{\lambda_1^B}, \dots, \sqrt{\lambda_d^B}\right).$$

*Then, the RISWIE distance (using all d PCA axes) admits the closed-form:*

$$D^2(A, B) = \frac{1}{d} \|\mathbf{a} - \mathbf{b}\|_2^2.$$

The square roots of the eigenvalues are standard deviations along a principal axis. This result is intuitive given that projecting a Gaussian distribution onto any vector yields another Gaussian.

**Theorem 4** (RISWIE–GW Comparison for Gaussians). *Let $A$ and $B$ satisfy the same assumptions as in Theorem 3 and additionally be full rank. Let $a_i := \sqrt{\lambda_i^A}$ and $b_i := \sqrt{\lambda_i^B}$, and define*

$$\alpha := \min_{1 \leq i \leq d}(a_i + b_i), \qquad \beta := \max_{1 \leq i \leq d}(a_i + b_i).$$

*Then the RISWIE distance under PCA embeddings satisfies:*

*(i) **Upper bounds.***

$$D^2(A, B) \ \leq \ \frac{GW_2^2(A, B)}{8d\,\alpha^2} \ + \ \frac{\|\Sigma_A\|_F \,\|\Sigma_B\|_F}{d\,\alpha^2}\left(1 - \frac{1}{\sqrt{d}}\right),$$

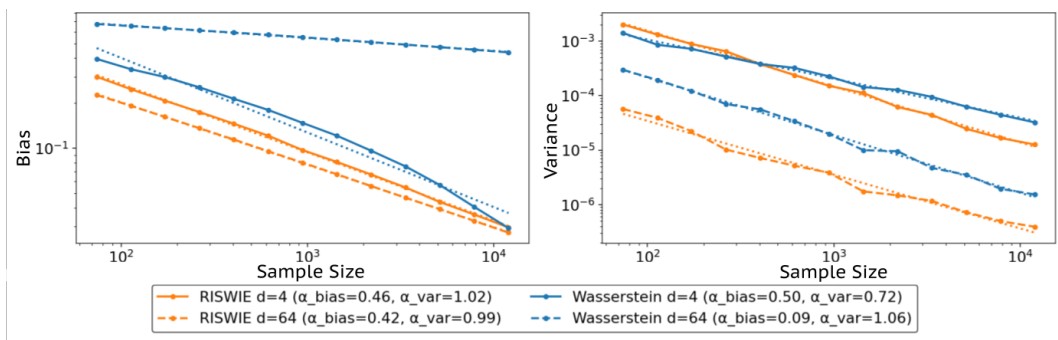

Figure 1: RISWIE-PCA vs. OT: bias (left) and variance (right). RISWIE bias and variance do not worsen in higher dimensions. Ground-truth population distances are calculated with the Gaussian closed form from both distances. The exponent $\alpha$ corresponds to the empirical decay rate in the log–log plot: we fit a power law of the form $An^{-\alpha}$ to each curve to estimate the convergence rate.

*and also*

$$D^2(A, B) \leq \frac{1}{2\sqrt{d}} \sqrt{GW_2^2(A, B) - 4\left(\operatorname{tr}(\Lambda_A) - \operatorname{tr}(\Lambda_B)\right)^2 - 4\left(\|\Lambda_A\|_F - \|\Lambda_B\|_F\right)^2}$$
$$\leq \frac{GW_2(A, B)}{2\sqrt{d}}.$$

*(ii)* ***Lower bound.***

$$D(A, B) \geq \frac{GW_2(A, B)}{2\,\beta\,\sqrt{d(d+2)}}.$$

Gromov-Wasserstein for Gaussians has no closed form, but there have been proven lower and upper bounds for it in the Gaussian case (Salmona et al., 2022). Interestingly, we were able to relate RISWIE$^2$ to both $GW_2$ and $GW_2^2$. The $\alpha$ normalization resolves the difference in units.

### 3.3 STATISTICAL PROPERTIES

As one may expect, under reasonable assumptions, $D(\hat{\mu}_n, \hat{\nu}_n) \xrightarrow{\text{a.s.}} D(\mu, \nu)$ as $n \to \infty$ where $\hat{\mu}_n, \hat{\nu}_n$ denote empirical measures of size $n$ drawn i.i.d. from $\mu, \nu$ (see Theorem 6 in the Appendix). However, a finite sample will always include bias. Consider $D(\mu, \mu) = 0$, yet $\mathbb{E}\left[D(\hat{\mu}_n, \hat{\mu}_n')\right] > 0$ where $\hat{\mu}_n'$ is an another independent empirical measure of size $n$ drawn i.i.d. from $\mu$. Thus, it is important to consider the bias and variance of $D(\hat{\mu}_n, \hat{\nu}_n)$.

Figure 1 empirically investigates the finite-sample convergence guarantees of the RISWIE-PCA and Wasserstein-2 distances relative to the population distance, which are made possible by the Gaussian closed-form that each distance has. We sample $n$ points from two Gaussian distributions repeatedly, recording the empirical distances between the resulting point clouds and comparing their average to the true population value (bias), as well as their sample variance across trials.

RISWIE exhibits strong empirical statistical behavior–for both low and high-dimensional settings, the bias scales as $O(n^{-1/2})$ and variance as $O(n^{-1})$. In contrast, $W_2$ converges with a rate of $O(n^{-1/d})$, meaning exponentially many samples are needed to get the same error as in lower dimensions (Weed & Bach, 2017). This is problematic given the computational cost associated with more samples for $W_2$ and similar distances.

## 4 EXPERIMENTS

We evaluated RISWIE with PCA embeddings in classification tasks, using the MPI-FAUST dataset of human meshes (Bogo et al., 2014) and spatially resolved tissue data from the HuBMAP consortium (Hickey et al., 2023). The below numerical results quantify computational efficiency and assess

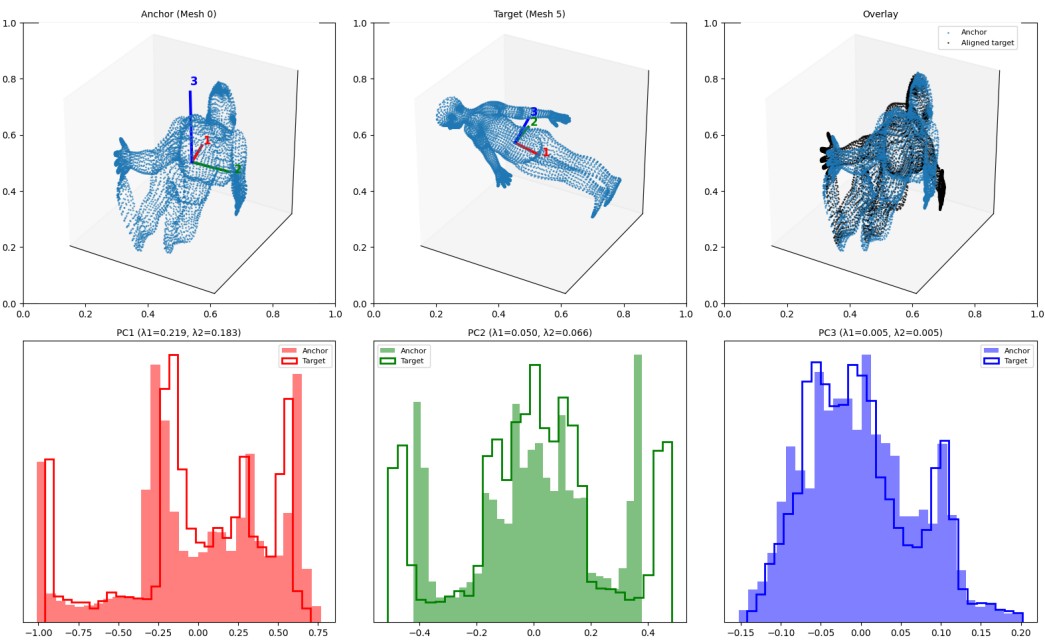

Figure 2: 3D Example of RISWIE alignment. We illustrate how RISWIE aligns two point clouds by matching their marginal distributions along embedded axes. This method naturally extends to higher dimensions. For each axis of the anchor shape, we evaluate all possible pairings with axes of the target, including sign flips (reflections) to minimize the 1D Wasserstein cost. The second row shows the optimal axis matching determined by this process, and we show the poses overlaid with this alignment procedure.

discriminative, clustering, and classification performance relative to existing distances. We use the Python Optimal Transport (POT) library's implementations of Gromov–Wasserstein (via an approximate solver) and Wasserstein (standard OT) in our comparisons (Flamary et al., 2021; 2024), and the Procrustes-Wasserstein implementation from Adamo et al. (2025). For FAUST and HuBMAP experiments, we sample 64 axes to ensure robustness against variability in sampling from the unit sphere.

## 4.1 HUMAN POSE ALIGNMENT AND DISCRIMINATION

On MPI-FAUST, we treat each registered mesh as a point cloud and compare pairs from the same subject under distinct pose and orientations. As shown in Figure 2, RISWIE aligns the target to the anchor by matching principal axes up to permutation and sign. After alignment, the point clouds overlay closely and their 1D marginals along the first three principal components nearly coincide, indicating robustness to rigid motions.

We further evaluate unsupervised pose clustering on MPI-FAUST (10 subjects × 10 poses). For each method, we compute a $100 \times 100$ pairwise distance matrix and embed each mesh as a row. For consistency, all distances are calculated with 1000 subsampled vertices per mesh. This is done for the computability of Wasserstein and Gromov-Wasserstein. However, RISWIE could use all 6890 vertices at negligible extra cost.

We evaluate K-Means, Spectral, Agglomerative, and t-SNE–based clustering on mesh embeddings (the row vectors of the distance matrix), measuring performance with V-measure, ARI, and accuracy. Table 1 reports V-measure: RISWIE matches or outperforms GW and other baselines across clustering strategies. Over our grid of settings, RISWIE surpasses GW in V-measure and NMI in 90.9% of cases and in ARI and accuracy in 100% of cases, while computing the full distance ma-

trix in ∼10 seconds versus ∼5 hours for GW. Thus, regardless of the clustering method used in unsupervised learning, RISWIE provides consistently strong and efficient performance.

Table 1: Mean V-measure (across subsampling with 10 different seeds) by method and distance function on MPI-FAUST pose clustering.

| Distance
Pipeline | Euclidean | Gromov | Wasserstein | Procrustes | RISWIE | Sliced |
|---|---|---|---|---|---|---|
| Agglomerative (avg, precomp) | 0.2214 | 0.6568 | 0.6715 | 0.6794 | **0.8094** | 0.5478 |
| KMeans (dist rows) | 0.3778 | 0.5930 | 0.5967 | 0.4918 | **0.7839** | 0.4331 |
| Spectral (RBF of dist) | 0.3721 | 0.5630 | 0.5757 | 0.5924 | **0.8138** | 0.6291 |
| t-SNE-2D + KMeans | 0.4066 | 0.6649 | 0.6480 | 0.7209 | **0.8612** | 0.6329 |
| t-SNE-2D + Spectral | 0.3907 | 0.6481 | 0.6136 | 0.6417 | **0.8196** | 0.6173 |
| AUC-ROC (same-vs-different) | 0.6099 | 0.8929 | 0.8603 | 0.8677 | **0.9404** | 0.7843 |

## 4.2    Tissue Clustering

We evaluate RISWIE on two-dimensional tissue slices of the human small intestine, where each slice is represented as a point cloud of cell coordinates (Hickey et al., 2023), orientated arbitrarily. Ground-truth labels group slices by intestine identity.

We compute the all-pairs RISWIE distance matrix between point clouds from different tissue types and vertical slices. Each block in the matrix compares all slices of one tissue to all slices of another. Since each slice may be arbitrarily rotated or reflected, a rigid-invariant distance should yield low pairwise values within diagonal blocks (same tissue), despite variations in orientation or sampling. Figure 3 highlights RISWIE's robustness to such transformations, showing consistently low intra-tissue distances.

Table 2 reports runtime and stack assignment accuracy across distances. For clustering/assignment, we apply a farthest-point seeding strategy with greedy assignment based on intra-cluster distances, with more information available in Appendix A.3. RISWIE achieves sub-second computation and the highest accuracy (95.8%), while Gromov–Wasserstein is slower by over four orders of magnitude. Sliced Wasserstein and classical Wasserstein are faster than GW but substantially less accurate.

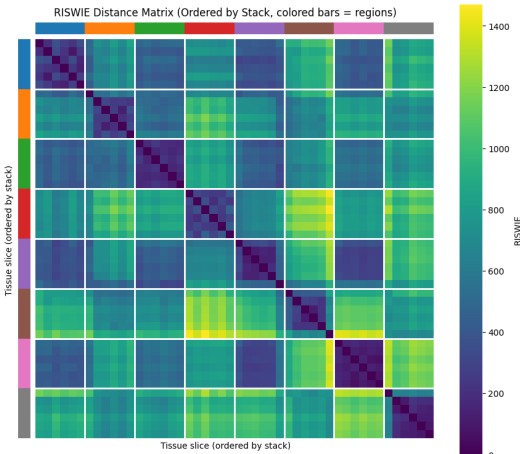

Figure 3: RISWIE Distance matrix for the HuBMAP tissue slices. Each block along the diagonal corresponds to slices from the same tissue stack. Within a block, RISWIE distances are consistently near zero, indicating strong invariance to small perturbations and local alignment of slices from the same sample.

| Distance | Time (s) | Accuracy |
|---|---|---|
| RISWIE | **1** | **95.83**% |
| Gromov–Wasserstein | 10352 | 85.42% |
| Procrustes–Wasserstein | 4952 | 77.08% |
| Sliced Wasserstein | 2 | 52.08% |
| Wasserstein | 111 | 54.17% |

Table 2: Cells dataset: runtime and stack assignment accuracy using 1000-point subsampling.

Beyond assignment, RISWIE provides stronger discriminative power. Using pairwise distances to score same-intestine versus different-intestine pairs, RISWIE achieves an AUC-ROC of 0.943 compared to 0.921 for Gromov–Wasserstein or 0.829 for Procrustes–Wasserstein (all under identical subsampling). Since RISWIE scales nearly linearly with sample size, it can exploit larger point sets with little additional cost, which would further improve discriminatory power. However, we again subsample the same number of points for consistency.

## 5 DISCUSSION

Our empirical results demonstrate that the massive computational benefits of RISWIE do not degrade accuracy – in fact, they can often improve it relative to approximate solvers. On both applications, RISWIE achieved higher AUC than approximating Gromov-Wasserstein or Procrustes-Wasserstein in 4 orders of magnitude less time. RISWIE also recovers a signed axis permutation between axes, which, when using PCA, can be used to give an explicit rotation/reflection aligning two shapes. As a result, we can define rigid-invariant versions of any distance function–apply RISWIE's alignment step and then evaluate the other distance. This makes RISWIE useful both as a standalone distance measure and as a preprocessing step for downstream geometric data analysis.

Two limitations should be noted. First, our method relies on discrete axis matchings, corresponding to optimization over the vertices of the (signed) Birkhoff polytope. Consequently, the objective is non-differentiable, which limits compatibility with gradient-based learning frameworks (Alvarez-Melis & Jaakkola, 2018). In Appendix A.6, we introduce a generalization that relaxes this vertex-level optimization to the interior of the Birkhoff polytope using entropic optimal transport, yielding probabilistic axis matchings. This formulation preserves rigid invariance, is fully differentiable, and recovers the original method as a limiting case of the regularization parameters.

Second, performance depends on embedding stability, since small eigengaps render individual embedding axes non-identifiable. RISWIE is generally robust to this ambiguity by optimizing over all signed permutations of the embedding basis (see Appendix A.5). Geometrically, RISWIE matches one-dimensional subspaces, corresponding to points on the Grassmannian $\mathrm{Gr}(1, d)$. A natural extension to handle $k$-dimensional degenerate eigenspaces is to treat the associated eigenspaces as higher-dimensional blocks and match them as elements of the Grassmannian $\mathrm{Gr}(k, d)$.

Together, these experimental and theoretical results position RISWIE as a practical tool for large-scale geometric data analysis and a foundation for future work on invariant transport methods.

## ACKNOWLEDGMENTS

This work was supported by the National Science Foundation under Grant DMS-2038056. The authors thank Jiajia Yu, Heekyoung Hahn, and Lenny Ng for their guidance and valuable feedback throughout the project.

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

# A    APPENDIX

## A.1    TIMING RESULTS

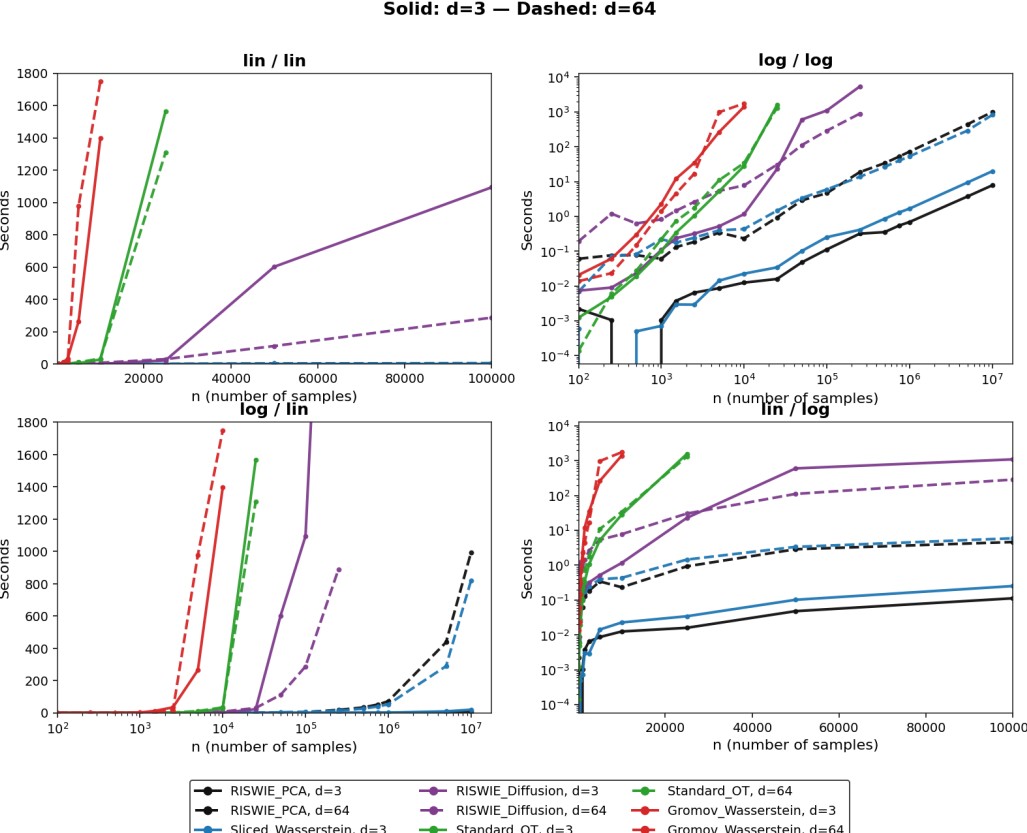

For our timing experiments, we set the number of projection axes for Sliced Wasserstein to $\max(10, d\log d)$ and the number of embedding functions of RISWIE-PCA to $d$. The former is done to make Sliced Wasserstein robust to bad sampling directions as they are not data dependent. For diffusion-based RISWIE, we implement diffusion maps by building a sparse neighborhood graph with $k = \lceil d\log n \rceil$ neighbors, then apply heat-kernel affinities and symmetric normalization before computing the top $d$ eigenvectors.

## A.2    FAUST FULL EXPERIMENT

Tables for this experiment include clustering pipelines, where abbreviations like "avg, precomp", "dist rows", and "RBF of dist" refer to specific clustering setups described in the table caption and glossary.

Table 3: Description of clustering pipelines used in the experiments.

| Pipeline Label | Description |
|---|---|
| KMeans (dist rows) | KMeans on rows of the pairwise distance matrix as Euclidean vectors. |
| KMedoids (precomputed dist) | KMedoids using the full precomputed pairwise distance matrix. |
| Agglomerative (avg, precomp) | Average-linkage agglomerative clustering on the precomputed distance matrix. |
| Spectral (RBF of dist) | Spectral clustering using an RBF kernel of the distance matrix: $A_{ij} = \exp\left(-D_{ij}^2/(2\sigma^2)\right)$ with $\sigma = \mathrm{median}(D[D > 0])$. |
| MDS-2D + KMeans | 2D MDS embedding of distances followed by KMeans. |
| MDS-3D + KMeans | 3D MDS embedding of distances followed by KMeans. |
| MDS-2D + Spectral | 2D MDS embedding, RBF kernel on embedded points, then Spectral clustering. |
| t-SNE-2D + KMeans | 2D t-SNE on precomputed distances (perplexity 10), then KMeans. |
| t-SNE-3D + KMeans | 3D t-SNE on precomputed distances, then KMeans. |
| t-SNE-2D + Spectral | 2D t-SNE followed by RBF kernel and Spectral clustering. |
| t-SNE-3D + Spectral | 3D t-SNE followed by RBF kernel and Spectral clustering. |

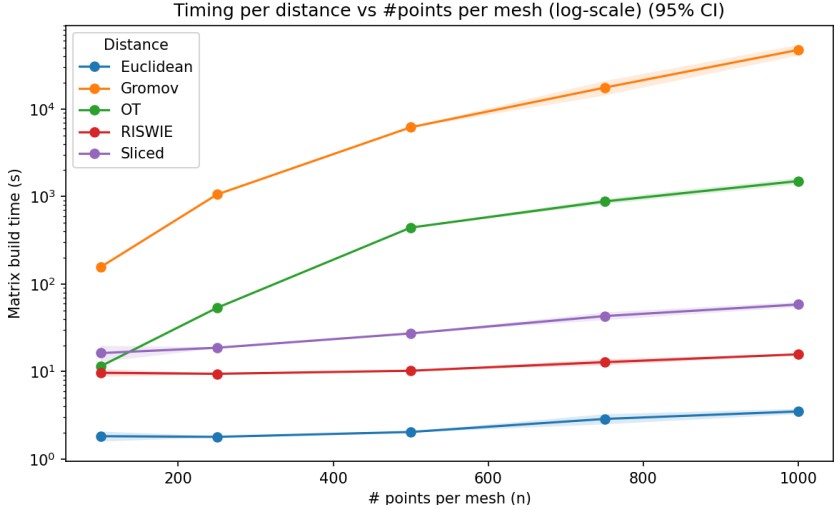

Figure 4: Matrix-build time versus number of points per mesh $n$ (log-scale). RISWIE grows gently with $n$ and stays well below Sliced/OT, while Gromov–Wasserstein is the slowest by far.

Table 4: Accuracy by Method and Distance Function

| Method | RISWIE | Gromov | OT | Euclidean | Sliced |
|---|---|---|---|---|---|
| KMeans (dist rows) | **0.7200** | 0.5700 | 0.5600 | 0.3500 | 0.3600 |
| Spectral (RBF of dist) | **0.7800** | 0.7500 | 0.5300 | 0.3200 | 0.6000 |
| Agglomerative (avg, precomp) | **0.7200** | 0.5300 | 0.4600 | 0.1400 | 0.4500 |
| MDS-2D + KMeans | **0.7300** | 0.5800 | 0.5400 | 0.3100 | 0.4200 |
| MDS-2D + Spectral | **0.5800** | 0.4600 | 0.4300 | 0.3200 | 0.3300 |
| MDS-3D + KMeans | **0.7800** | 0.7000 | 0.5000 | 0.3200 | 0.4300 |
| MDS-3D + Spectral | **0.7300** | 0.6700 | 0.5200 | 0.3100 | 0.4200 |
| t-SNE-2D + KMeans | **0.8700** | 0.8200 | 0.6500 | 0.4100 | 0.6100 |
| t-SNE-2D + Spectral | **0.7200** | 0.6800 | 0.5600 | 0.4100 | 0.5300 |
| t-SNE-3D + KMeans | **0.8000** | 0.7500 | 0.5300 | 0.3500 | 0.5200 |
| t-SNE-3D + Spectral | **0.7600** | 0.6800 | 0.5700 | 0.3000 | 0.5000 |

Table 5: V-measure by Method and Distance Function

| Method | RISWIE | Gromov | OT | Euclidean | Sliced |
|---|---|---|---|---|---|
| KMeans (dist rows) | **0.8058** | 0.6802 | 0.5957 | 0.4007 | 0.4373 |
| Spectral (RBF of dist) | 0.8238 | **0.8303** | 0.5790 | 0.3220 | 0.6437 |
| Agglomerative (avg, precomp) | **0.8082** | 0.7420 | 0.6763 | 0.2137 | 0.6092 |
| MDS-2D + KMeans | **0.7454** | 0.6721 | 0.5506 | 0.2986 | 0.4386 |
| MDS-2D + Spectral | **0.7065** | 0.5958 | 0.4921 | 0.3161 | 0.3510 |
| MDS-3D + KMeans | **0.8231** | 0.7879 | 0.5818 | 0.2870 | 0.4892 |
| MDS-3D + Spectral | **0.7789** | 0.7422 | 0.5700 | 0.3162 | 0.4676 |
| t-SNE-2D + KMeans | **0.8829** | 0.8577 | 0.6779 | 0.4138 | 0.6246 |
| t-SNE-2D + Spectral | **0.8291** | 0.7896 | 0.6357 | 0.3954 | 0.6022 |
| t-SNE-3D + KMeans | **0.7832** | 0.7606 | 0.5847 | 0.3486 | 0.5281 |
| t-SNE-3D + Spectral | **0.7754** | 0.7039 | 0.5843 | 0.2856 | 0.4686 |

Table 6: Adjusted Rand Index (ARI) by Method and Distance Function

| Method | RISWIE | Gromov | OT | Euclidean | Sliced |
|---|---|---|---|---|---|
| KMeans (dist rows) | **0.5844** | 0.3910 | 0.3673 | 0.1359 | 0.1618 |
| Spectral (RBF of dist) | **0.6825** | 0.6154 | 0.3312 | 0.0944 | 0.4277 |
| Agglomerative (avg, precomp) | **0.5526** | 0.4197 | 0.3796 | 0.0171 | 0.3498 |
| MDS-2D + KMeans | **0.5454** | 0.3906 | 0.3067 | 0.0486 | 0.1723 |
| MDS-2D + Spectral | **0.4363** | 0.2881 | 0.2318 | 0.0696 | 0.1078 |
| MDS-3D + KMeans | **0.6531** | 0.5645 | 0.3336 | 0.0499 | 0.2214 |
| MDS-3D + Spectral | **0.5576** | 0.5028 | 0.3427 | 0.0732 | 0.2026 |
| t-SNE-2D + KMeans | **0.7965** | 0.7416 | 0.4946 | 0.1765 | 0.4116 |
| t-SNE-2D + Spectral | **0.6436** | 0.5718 | 0.4102 | 0.1480 | 0.3569 |
| t-SNE-3D + KMeans | **0.6529** | 0.6085 | 0.3552 | 0.1013 | 0.3136 |
| t-SNE-3D + Spectral | **0.6107** | 0.4572 | 0.3301 | 0.0584 | 0.2254 |

A.3   CELLS FULL EXPERIMENT

To evaluate RISWIE's effectiveness in recovering biologically meaningful groupings, we perform balanced partitioning of tissue slices into spatial stacks based on the computed pairwise distances between tissue slices. We use a farthest-point seeding strategy to encourage diversity among initial stack centers and apply a greedy assignment procedure to add tissue slices to a cluster that they are most similar to.

In other words, we are trying to minimize

$$\mathcal{L}(\mathcal{S}_1, \ldots, \mathcal{S}_K) = \sum_{k=1}^{K} \sum_{\substack{i,j \in \mathcal{S}_k \\ i<j}} D_{\text{Input Distance}}(X_i, X_j)$$

where $\mathcal{X} = \{X_1, X_2, \ldots, X_n\}$ is the set of tissue slices and we want to partition them into stacks $\mathcal{S}_1, \ldots, \mathcal{S}_K$, each of size $n/K$.

Table 7: Normalized Mutual Information (NMI) by Method and Distance Function

| Method | RISWIE | Gromov | OT | Euclidean | Sliced |
|---|---|---|---|---|---|
| KMeans (dist rows) | **0.8058** | 0.6802 | 0.5957 | 0.4007 | 0.4373 |
| Spectral (RBF of dist) | 0.8238 | **0.8303** | 0.5790 | 0.3220 | 0.6437 |
| Agglomerative (avg, precomp) | **0.8082** | 0.7420 | 0.6763 | 0.2137 | 0.6092 |
| MDS-2D + KMeans | **0.7454** | 0.6721 | 0.5506 | 0.2986 | 0.4386 |
| MDS-2D + Spectral | **0.7065** | 0.5958 | 0.4921 | 0.3161 | 0.3510 |
| MDS-3D + KMeans | **0.8231** | 0.7879 | 0.5818 | 0.2870 | 0.4892 |
| MDS-3D + Spectral | **0.7789** | 0.7422 | 0.5700 | 0.3162 | 0.4676 |
| t-SNE-2D + KMeans | **0.8829** | 0.8577 | 0.6779 | 0.4138 | 0.6246 |
| t-SNE-2D + Spectral | **0.8291** | 0.7896 | 0.6357 | 0.3954 | 0.6022 |
| t-SNE-3D + KMeans | **0.7832** | 0.7606 | 0.5847 | 0.3486 | 0.5281 |
| t-SNE-3D + Spectral | **0.7754** | 0.7039 | 0.5843 | 0.2856 | 0.4686 |

Table 8: Clustering performance using RISWIE with no subsampling. Accuracy, V-measure, ARI, and NMI are reported across clustering pipelines.

| Method | Accuracy | V-measure | ARI | NMI |
|---|---|---|---|---|
| KMeans (dist rows) | 0.7500 | 0.8469 | 0.6446 | 0.8469 |
| KMedoids (precomputed dist) | 0.8200 | 0.8296 | 0.6966 | 0.8296 |
| Spectral (RBF of dist) | 0.7900 | 0.8343 | 0.6921 | 0.8343 |
| Agglomerative (avg, precomp) | 0.7800 | 0.8549 | 0.6655 | 0.8549 |
| MDS-2D + KMeans | 0.7500 | 0.7756 | 0.5934 | 0.7756 |
| MDS-2D + KMedoids | 0.7500 | 0.7666 | 0.5878 | 0.7666 |
| MDS-2D + Spectral | 0.6600 | 0.7531 | 0.5121 | 0.7531 |
| MDS-3D + KMeans | 0.7300 | 0.7517 | 0.5608 | 0.7517 |
| MDS-3D + KMedoids | 0.7100 | 0.7541 | 0.5776 | 0.7541 |
| MDS-3D + Spectral | 0.7200 | 0.7843 | 0.5382 | 0.7843 |
| t-SNE-2D + KMeans | 0.8300 | 0.8498 | 0.7348 | 0.8498 |
| t-SNE-2D + KMedoids | 0.8300 | 0.8498 | 0.7348 | 0.8498 |
| t-SNE-2D + Spectral | 0.7000 | 0.8339 | 0.6081 | 0.8339 |
| t-SNE-3D + KMeans | 0.7600 | 0.7850 | 0.6276 | 0.7850 |
| t-SNE-3D + KMedoids | 0.7700 | 0.7633 | 0.6116 | 0.7633 |
| t-SNE-3D + Spectral | 0.6400 | 0.7145 | 0.4688 | 0.7145 |

---

**Algorithm 2:** Stack Assignment via RISWIE, Farthest-Point Seeding, and Greedy Assignment

**Input:** Set of $n = 48$ regions (point clouds) $\{X_i\}$
**Output:** Optimal grouping of regions into $K$ balanced stacks
**Step 1: Compute Distance Matrix**
**for** $i = 1$ *to* $n$ **do**
    **for** $j = i + 1$ *to* $n$ **do**
        $D_{ij} \leftarrow$ RISWIE_distance$(X_i, X_j)$ ;
        $D_{ji} \leftarrow D_{ij}$ ;

**Step 2: Farthest Point Seeding and Greedy Assignment**
**for** $s = 1$ *to* $n$        // Try each region as first seed **do**
    $S \leftarrow [s]$                   // Seed indices
    **while** $|S| < K$ **do**
        Select $t = \arg\max_{t \notin S} \min_{u \in S} D_{tu}$ ;
        Append $t$ to $S$ ;

    Initialize $K$ stacks, each with one seed from $S$ ;
    **while** *unassigned regions remain* **do**
        **for** *each unassigned region $r$, and each stack $k$ not full* **do**
            Compute cost $c_{r,k} = \sum_{b \in \text{stack}_k} D_{r,b}$ ;
        Assign $r^*$ to stack $k^*$ minimizing $c_{r,k}$, breaking ties arbitrarily ;

    Compute total within-stack sum $C_s = \sum_{k=1}^{K} \sum_{i,j \in S_k,\ i<j} D_{ij}$ ;
    Store stacks and $C_s$ ;

Select the stack assignment with lowest within-stack sum, summed across all stacks: $\sum_s C_s$ ;
**Step 3 (Optional): Random Seeds**
Optionally repeat the greedy assignment with some number of random initializations of $K$
  stacks and take the lowest cost stack assignment across all completed stacks.

Table 9: V-measure (mean ± std) by method and distance function on MPI-FAUST pose clustering.

| Distance Pipeline | Euclidean | Gromov | OT | RISWIE | Sliced |
|---|---|---|---|---|---|
| Agglomerative (avg, precomp) | 0.2214 ± 0.0252 | 0.6568 ± 0.0586 | 0.6715 ± 0.0164 | **0.8094 ± 0.0268** | 0.5478 ± 0.0346 |
| KMeans (dist rows) | 0.3778 ± 0.0257 | 0.5930 ± 0.0478 | 0.5967 ± 0.0259 | **0.7839 ± 0.0192** | 0.4331 ± 0.0292 |
| Spectral (RBF of dist) | 0.3721 ± 0.0248 | 0.5630 ± 0.0412 | 0.5757 ± 0.0225 | **0.8138 ± 0.0190** | 0.6291 ± 0.0387 |
| t-SNE-2D + KMeans | 0.4066 ± 0.0274 | 0.6649 ± 0.0447 | 0.6480 ± 0.0264 | **0.8612 ± 0.0270** | 0.6329 ± 0.0351 |
| t-SNE-2D + Spectral | 0.3907 ± 0.0308 | 0.6481 ± 0.0482 | 0.6136 ± 0.0215 | **0.8196 ± 0.0183** | 0.6173 ± 0.0275 |

The assignment accuracy reported reflects the best label alignment between predicted and ground truth stacks, computed via Hungarian matching.

RISWIE Aligned: Each stack aligned to its first region (column 1)

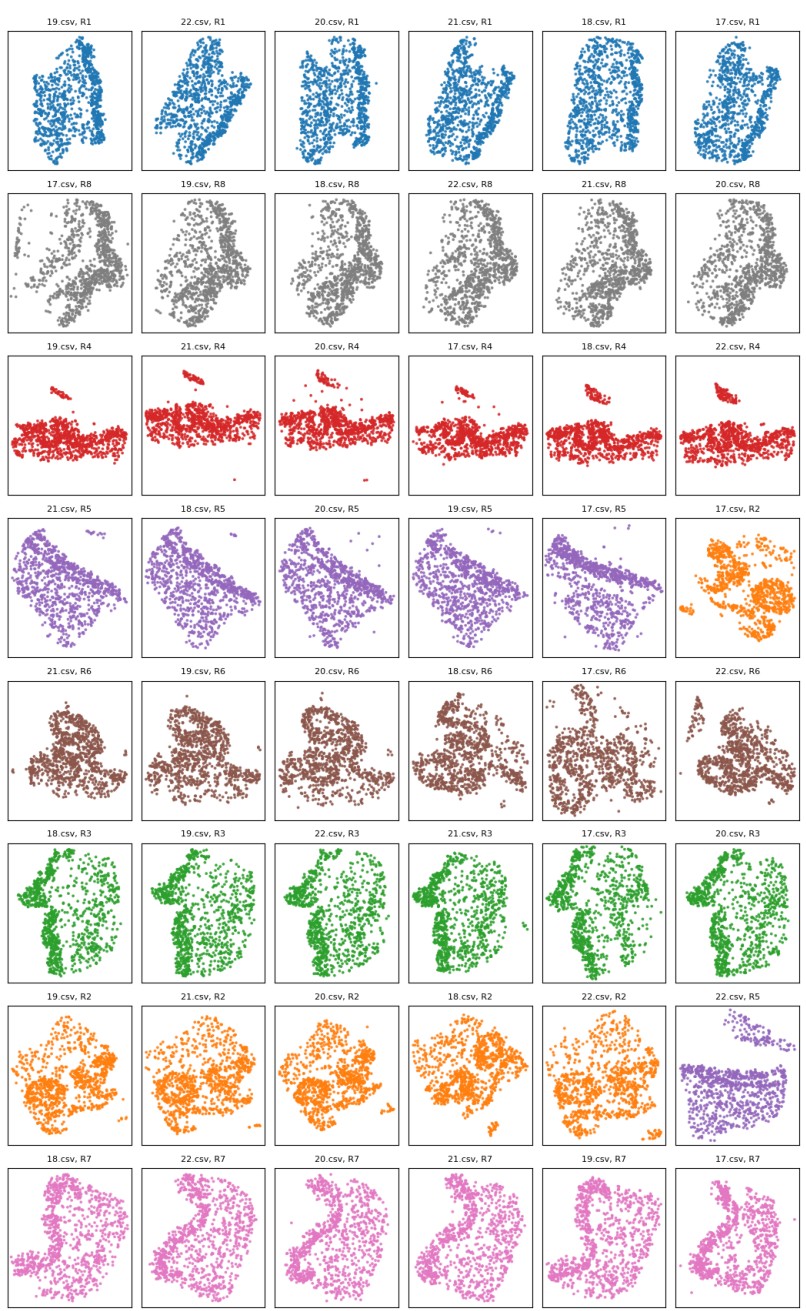
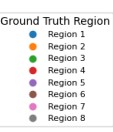

sliced Assignments (Each row = stack, cols = rotated tissues in that stack)
Color = true region

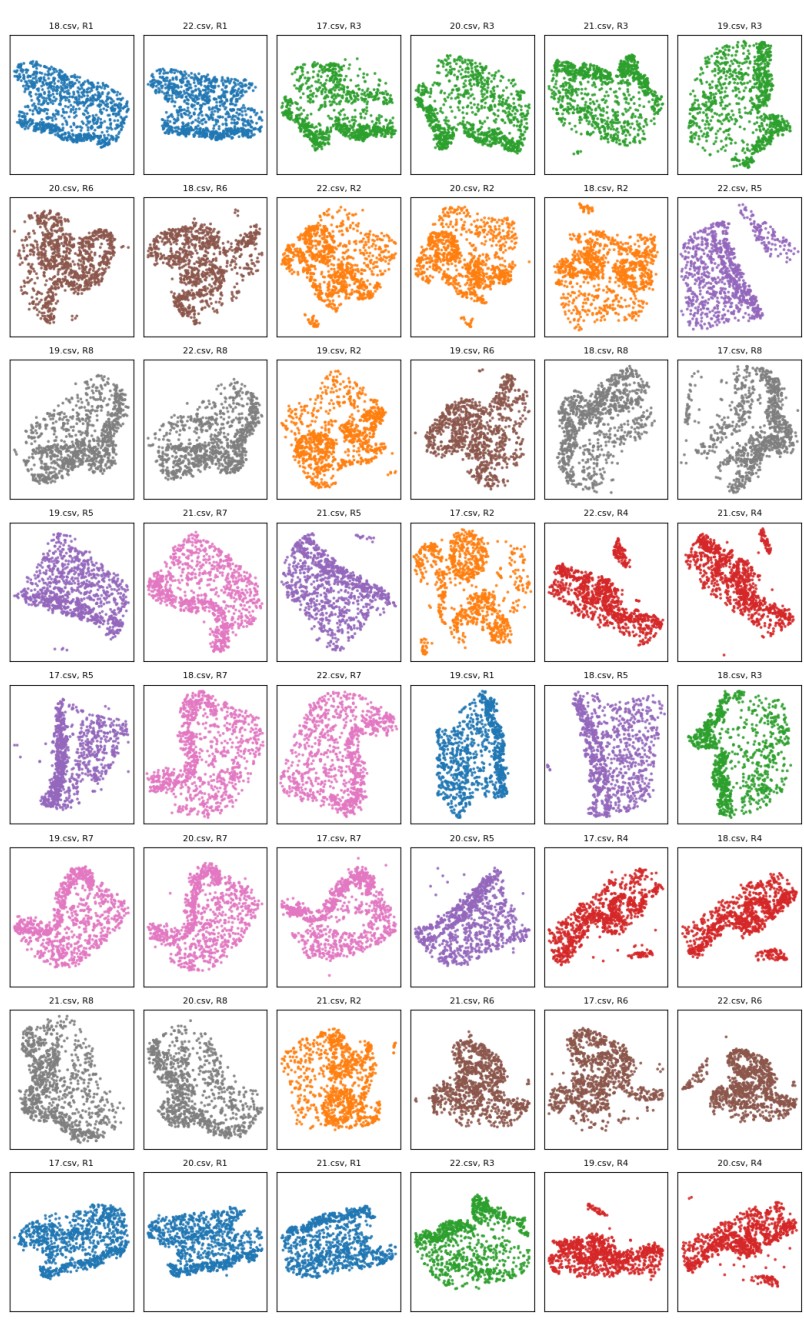

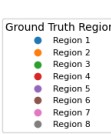

BOOSTED_SLICED Aligned: Each stack aligned to its first region (column 1)

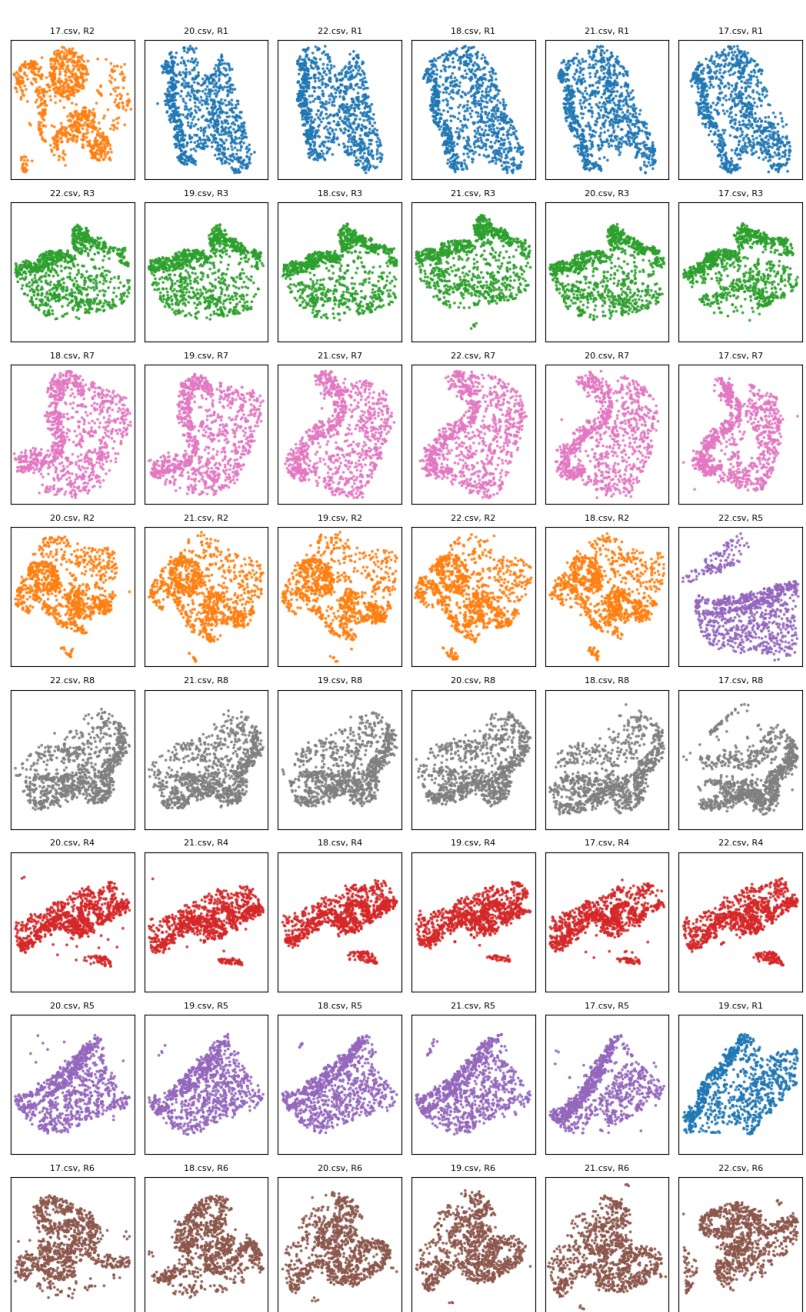
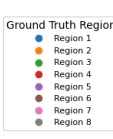

ot Assignments (Each row = stack, cols = rotated tissues in that stack)
Color = true region

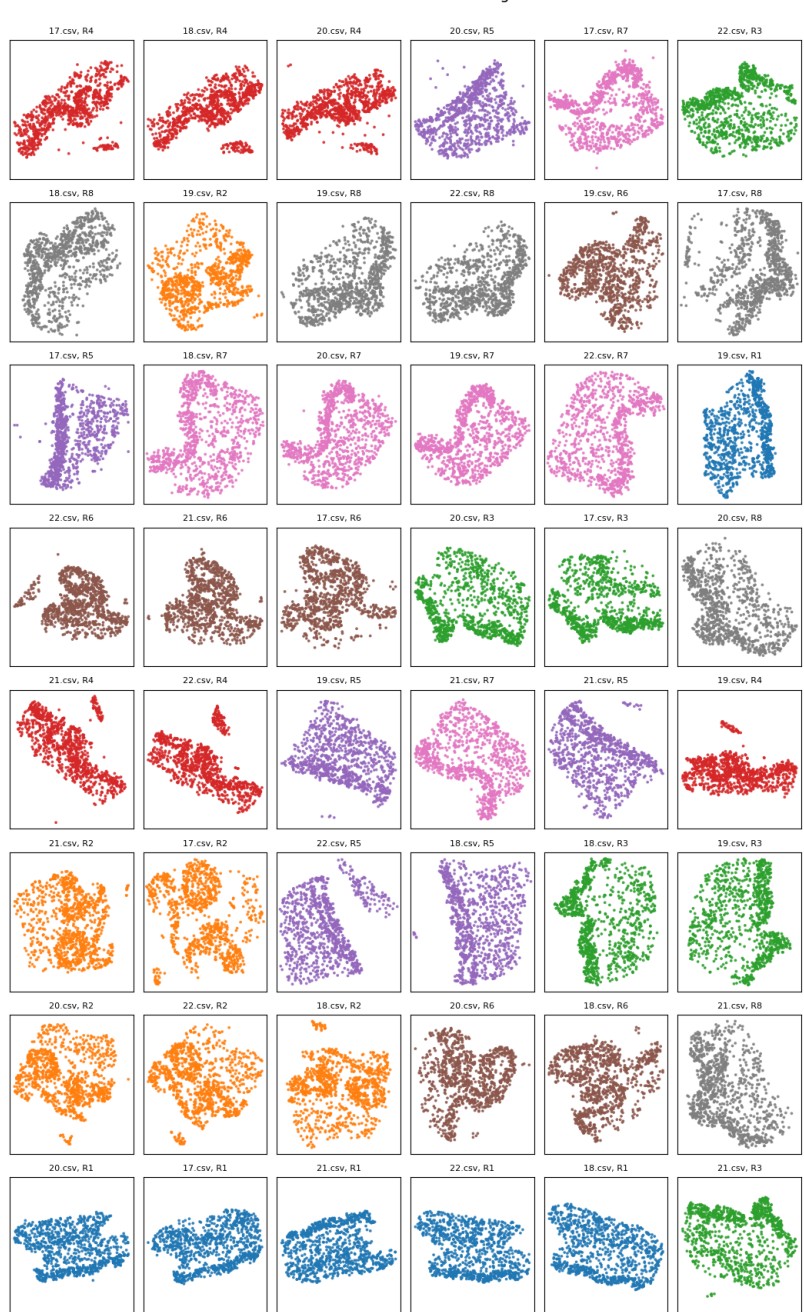

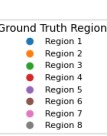

BOOSTED_OT Aligned: Each stack aligned to its first region (column 1)

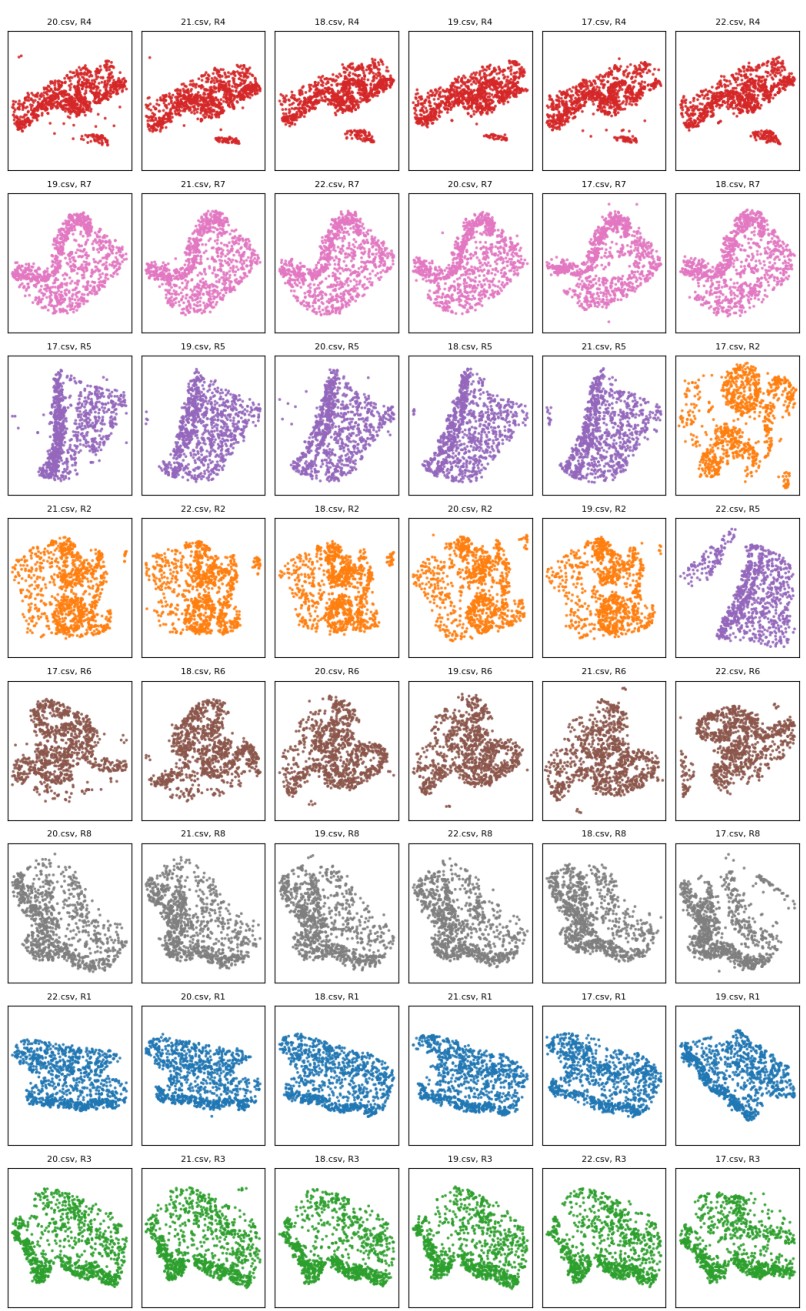
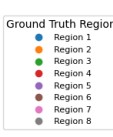

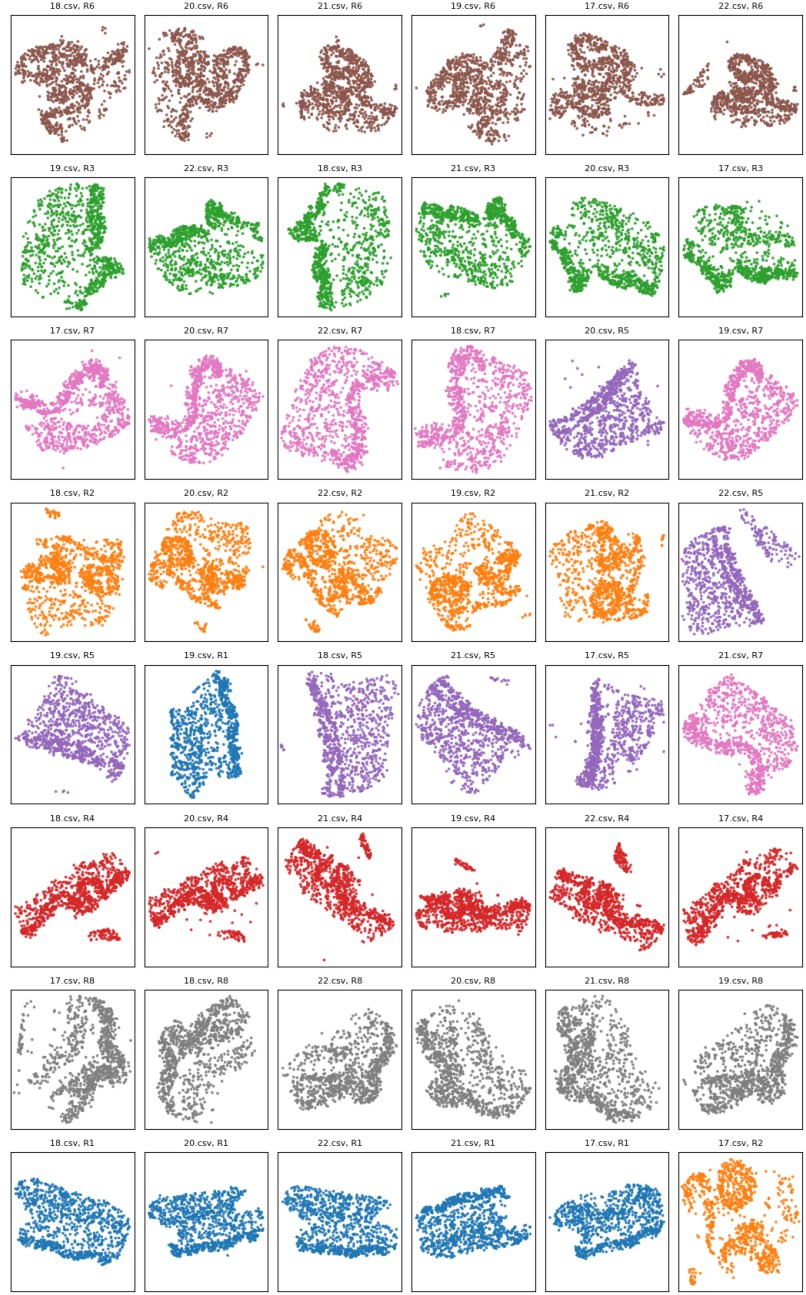
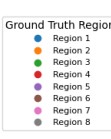

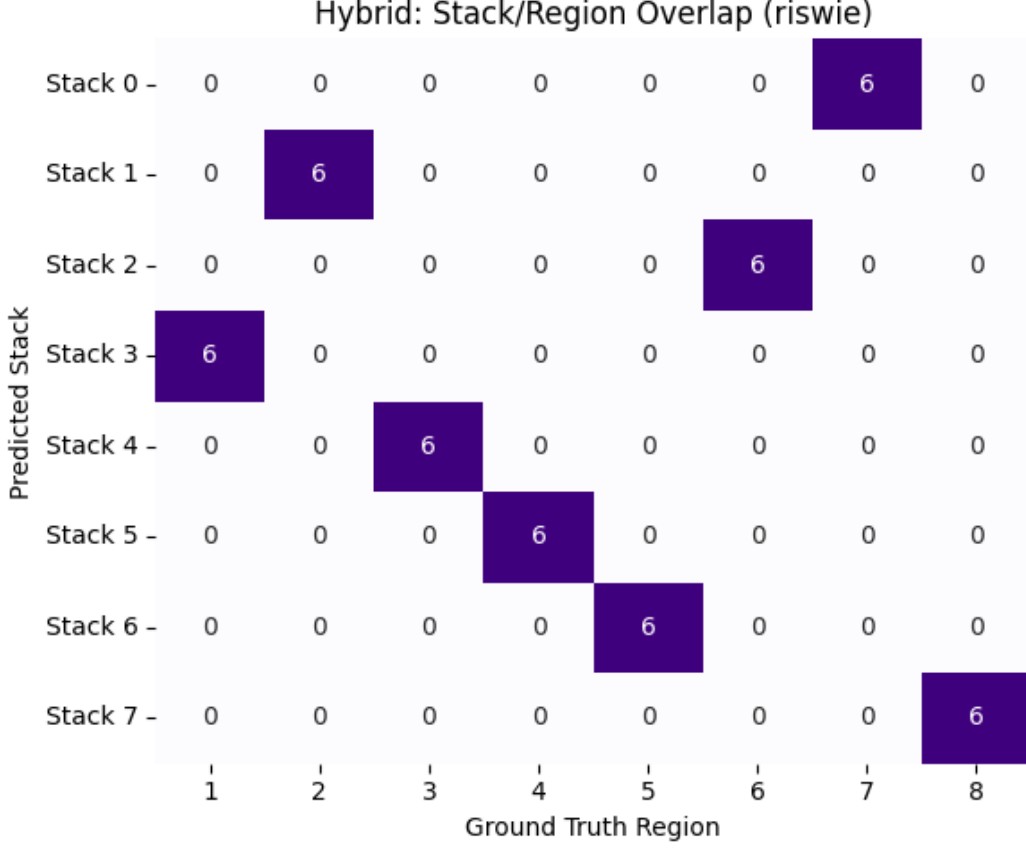

Figure 5: Hybrid Chosen Stack Assignment with RISWIE as the spatial distance and $\lambda = 0.5$

### A.3.1 HYBRID SPATIAL–MARKER DISTANCE AND STACK ASSIGNMENT

To incorporate both spatial structure and marker expression in our region-level comparisons, and taking inspiration from Vayer et al. (2019), we define a hybrid distance matrix that interpolates between them.

For each pair of regions, we compute two quantities.

- A spatial distance using a selected geometric distance function (e.g., RISWIE, etc), applied to the cell coordinates within each region.

- A marker distance computed as the 2-Wasserstein distance between high-dimensional cell marker embeddings sampled from each region.

Let $D_{ij}^{\text{spatial}}$ and $D_{ij}^{\text{marker}}$ denote these pairwise dissimilarities, both scaled to [0, 1] via min-max normalization.

We then define

$$D_{ij}^{\text{hybrid}} = \lambda \cdot D_{ij}^{\text{spatial}} + (1 - \lambda) \cdot D_{ij}^{\text{marker}},$$

where $\lambda \in [0, 1]$ is tunable.

We then use this hybrid distance matrix to perform stack assignment as before. Interestingly, $\lambda = 0.5$ is able to recover perfect stack accuracy using RISWIE as the spatial distance, while $\lambda = 1.0$ and $\lambda = 0.0$ were unable to.

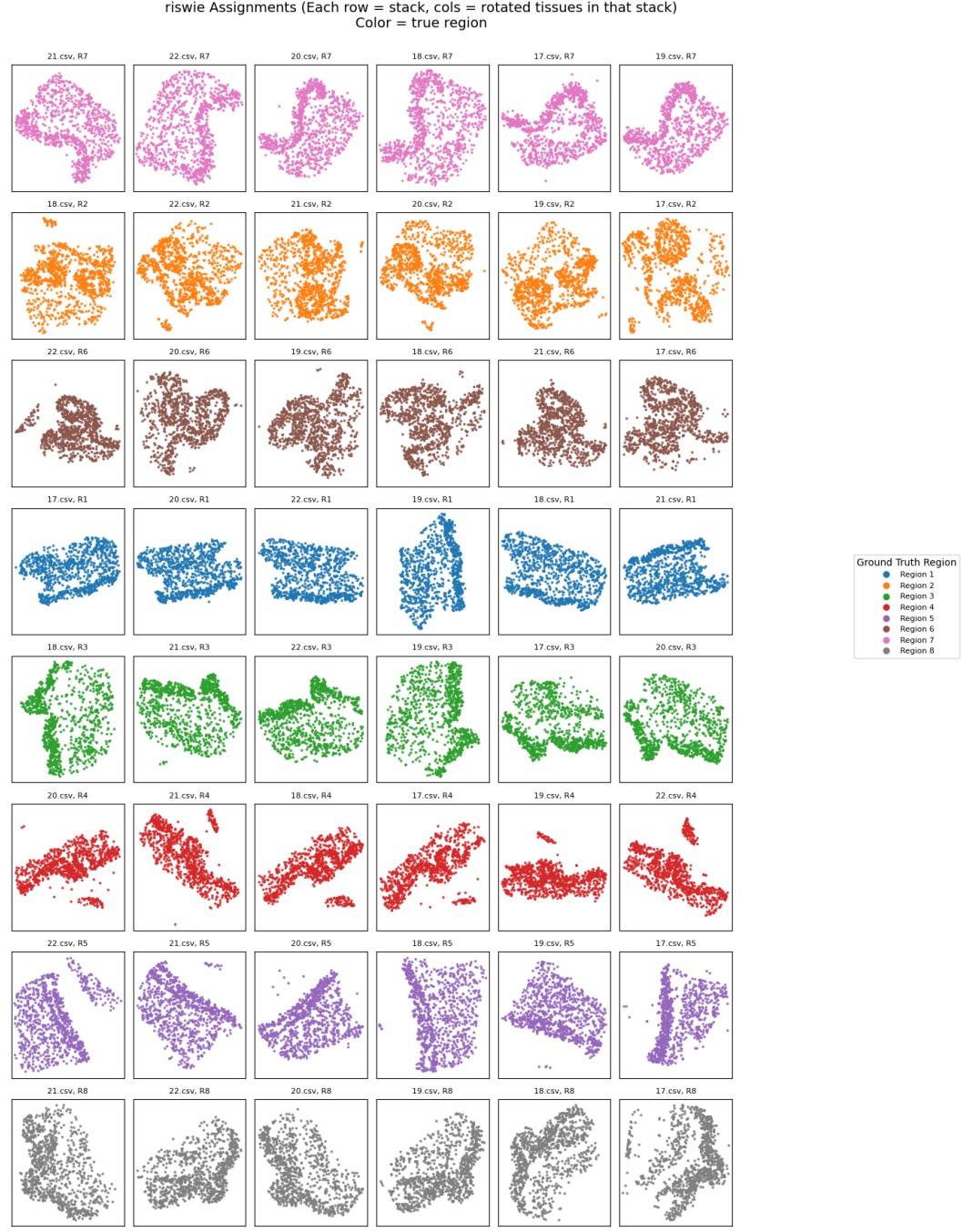

Figure 6: Unaligned Chosen Stack Assignment with RISWIE as the spatial distance and $\lambda = 0.5$

## A.4 Ordering Agreement Between RISWIE and Gromov–Wasserstein

We also investigate how often the ordering induced by Gromov–Wasserstein aligns with that induced by RISWIE. Specifically, for the cell dataset, we compute the proportion of consistent orderings:

$$\frac{\sum \mathbb{I}\left[\mathrm{sign}(\mathrm{GW}(a,b) - \mathrm{GW}(c,d)) = \mathrm{sign}(D(a,b) - D(c,d))\right]}{\sum 1}$$

where the sum ranges over all unique pairs of upper-triangular (off-diagonal) entries in the pairwise distance matrix.

Gromov–Wasserstein and RISWIE agreed on the ordering of 87.4% of all 635,628 region pair comparisons. The mean (median) absolute percentile difference between the two metrics was 0.091 (0.064).

When restricting to region pairs separated by at least one Gromov–Wasserstein standard deviation, the ordering agreement increased to 99.4% (302,853 out of 304,720 pairs).

Note that we approximate Gromov–Wasserstein using the solver provided in the POT library (Flamary et al., 2021; 2024). This does not guarantee exact agreement with the theoretical (NP-hard) Gromov–Wasserstein value.

## A.5 Near-Degenerate Eigenvalues

A natural concern for RISWIE is robustness when embedding coordinates are ambiguous, e.g., under (near-)degenerate spectra where eigenvectors may rotate within eigenspaces or swap order under small perturbations. RISWIE mitigates this by *axis matching* (via 1D Wasserstein distances) rather than relying on a fixed coordinate ordering.

**Degenerate eigenvalues experiment** We generate ten independent 500-point samples from each of eight highly symmetric shapes (sphere, cube, rectangular prism, pyramid, cylinder, torus, ellipsoid, and an L-shape), all on the same scale, and add isotropic Gaussian noise with standard deviation 0.02 independently to each point. We compute pairwise distance matrices and evaluate both pairwise discrimination (AUC) and clustering (V-measure). Table 10 shows that RISWIE outperforms rigid-invariant baselines (Procrustes–Wasserstein (PW) and we also compare to RISGW from Vayer et al. (2022)) across all downstream metrics. Additionally, on a reduced setting compatible with Gromov–Wasserstein, RISWIE again outperforms GW (Table 11).

Table 10: Degenerate-eigenvalue supporting experiment (mean $\pm$ std over runs).

| Metric | PW | RISGW | RISWIE |
|---|---|---|---|
| Pairwise AUC (same vs different) | $0.870 \pm 0.007$ | $0.742 \pm 0.007$ | $\mathbf{0.898 \pm 0.005}$ |
| KMeans on distance rows (V-measure) | $0.704 \pm 0.024$ | $0.540 \pm 0.036$ | $\mathbf{0.725 \pm 0.023}$ |
| Spectral (RBF of distance, V-measure) | $0.599 \pm 0.030$ | $0.236 \pm 0.052$ | $\mathbf{0.751 \pm 0.023}$ |
| Agglomerative (avg linkage, V-measure) | $0.655 \pm 0.030$ | $0.355 \pm 0.024$ | $\mathbf{0.666 \pm 0.032}$ |

Table 11: Compatibility run with fewer instances per shape to enable Gromov–Wasserstein (means shown).

| Method | AUC | KMeans (V) | Spectral (V) | Agglomerative (V) |
|---|---|---|---|---|
| Gromov–Wasserstein | 0.8347 | 0.6605 | 0.6411 | 0.5497 |
| RISWIE | **0.8960** | **0.7680** | **0.7900** | **0.6980** |

**Near-degenerate eigenvalues and noise sensitivity.** We next assess sensitivity to additional isotropic noise. Using the same setup, we report downstream performance both with no additional noise (beyond sampling noise) and with added Gaussian noise of standard deviation 0.05 (Table 12). RISWIE is largely unchanged, while PW exhibits larger shifts on several clustering metrics. We

also directly compare distance-matrix stability by correlating matrix entries between the no-noise and noisy conditions, and reporting mean relative error (MRE) (Table 13). Across noise levels, RISWIE-PCA is not more sensitive than Procrustes–Wasserstein.

Table 12: Near-degenerate experiment: downstream metrics (mean $\pm$ std). Top: no extra noise; bottom: added isotropic noise with $\sigma = 0.05$.

| Metric | PW | RISGW | RISWIE |
|---|---|---|---|
| AUC | $0.856 \pm 0.009$ | $0.770 \pm 0.005$ | $\mathbf{0.896 \pm 0.010}$ |
| KMeans V | $0.615 \pm 0.037$ | $0.520 \pm 0.040$ | $\mathbf{0.768 \pm 0.028}$ |
| Spectral V | $0.611 \pm 0.067$ | $0.349 \pm 0.055$ | $\mathbf{0.730 \pm 0.036}$ |
| Agglo V | $0.660 \pm 0.039$ | $0.467 \pm 0.059$ | $\mathbf{0.698 \pm 0.027}$ |
| AUC ($\sigma=0.05$) | $0.857 \pm 0.008$ | $0.770 \pm 0.005$ | $\mathbf{0.895 \pm 0.010}$ |
| KMeans V ($\sigma=0.05$) | $0.674 \pm 0.037$ | $0.584 \pm 0.030$ | $\mathbf{0.768 \pm 0.028}$ |
| Spectral V ($\sigma=0.05$) | $0.632 \pm 0.020$ | $0.348 \pm 0.056$ | $\mathbf{0.789 \pm 0.033}$ |
| Agglo V ($\sigma=0.05$) | $0.637 \pm 0.036$ | $0.467 \pm 0.059$ | $\mathbf{0.696 \pm 0.032}$ |

Table 13: Distance-matrix stability under added isotropic noise: correlation of distance-matrix entries and mean relative error (MRE) comparing no-noise vs noisy shapes.

| Noise $\sigma$ | RISWIE-PCA corr | RISWIE-PCA MRE | Procrustes corr | Procrustes MRE |
|---|---|---|---|---|
| 0.05 | 0.995 | 0.053 | 0.996 | 0.039 |
| 0.10 | 0.985 | 0.096 | 0.984 | 0.077 |
| 0.20 | 0.958 | 0.172 | 0.940 | 0.159 |
| 0.30 | 0.923 | 0.228 | 0.876 | 0.255 |

## A.6 RISWIE VARIANTS

To facilitate differentiable optimization, we define a soft relaxation of RISWIE, denoted SRISWIE, which replaces hard axis matching with entropic transport over a soft cost matrix. This provides a continuous approximation that is always rigid invariant and converges to RISWIE in the limit as $\beta \to \infty$ and $\varepsilon \to 0$ (the parameters $\beta$ and $\varepsilon$ control distinct relaxations).

**Definition 3** (Soft RISWIE (SRISWIE) Distance). Let $\mu, \nu$ be centered probability measures in $\mathcal{P}_2(\mathbb{R}^d)$, and again let $\varphi = (\varphi_1, \ldots, \varphi_k)$, $\psi = (\psi_1, \ldots, \psi_k)$ be fixed embedding functions.

For each $(j, m) \in \{1, \ldots, k\}^2$, define

$$C_{jm}^+ := W_2^2((\varphi_j)_\# \mu, \ (\psi_m)_\# \nu), \quad C_{jm}^- := W_2^2((\varphi_j)_\# \mu, \ (-\psi_m)_\# \nu)$$

and set the cost of a pairing as:

$$\tilde{C}_{jm} := w_{jm} C_{jm}^+ + (1 - w_{jm}) C_{jm}^-, \quad \text{where} \quad w_{jm} := \frac{1}{1 + \exp\left(\beta(C_{jm}^+ - C_{jm}^-)\right)}.$$

Let $\tilde{C} \in \mathbb{R}^{k \times k}$ be the resulting soft cost matrix. Define the SRISWIE distance as:

$$\text{SRISWIE}^2(\mu, \nu; \varepsilon, \beta) = \min_{\mathbf{P} \in \mathcal{U}_k} \left\{ \frac{1}{k} \sum_{j=1}^{k} \sum_{m=1}^{k} \mathbf{P}_{jm} \tilde{C}_{jm} + \varepsilon \sum_{j=1}^{k} \sum_{m=1}^{k} \mathbf{P}_{jm} \log \mathbf{P}_{jm} \right\}$$

where $\mathcal{U}_k$ denotes the set of $k \times k$ doubly stochastic matrices.

This variant replaces the hard signed-permutation matching over $O_k^\pm$ with an entropic optimal transport problem and handles axis reflections with a smooth soft-min. Geometrically, it induces a probability distribution over matchings between one-dimensional subspaces, corresponding to points on the Grassmannian $\text{Gr}(1, d)$. This formulation is easy to solve with Sinkhorn updates (analogously to entropically regularized Wasserstein distance).

We leave performance of SRISWIE on more sophisticated deep learning tasks for future work. On the FAUST dataset clustering task, SRISWIE was able to compute a $100 \times 100$ distance matrix between meshes with the full 6890 points in 34 seconds. Downstream spectral clustering with each mesh embedded as the corresponding row/column of the distance matrix yielded a V-measure of 0.8541.

We can also extract the optimal axis pairing and optimal relative sign for each axis pairing from RISWIE to align shapes before computing other distances such as Wasserstein or Sliced Wasserstein. We call these distances Boosted Optimal Transport and Boosted Sliced Wasserstein, respectively. See Section A.3 for comparisons of how these boosted distances perform in solving the balanced partitioning problem.

### A.7    THEOREMS AND PROOFS

Throughout the appendix, we will denote the RISWIE distance by $D$ unless stated otherwise (such as for the Gaussian closed form, denoted $D_G$).

*Proof of Theorem 1.* RISWIE is defined on centered embeddings (the means are subtracted), so translation $t$ has no effect on the pushforwards; we may assume $t = 0$ w.l.o.g.

**PCA:**    Let $\Sigma_\mu = U \Lambda U^\top$ be the eigendecomposition of the covariance where $\Lambda = \text{diag}(\lambda_1, \ldots, \lambda_d)$ and the eigenvalues are ordered $\lambda_1 > \cdots > \lambda_r > 0 = \lambda_{r+1} = \cdots = \lambda_d$

Applying $T(x) = Rx + t$, the covariance of $T_{\#}\mu$ is

$$\Sigma_{T_{\#}\mu} = R\Sigma_\mu R^\top = (RU)\Lambda(RU)^\top$$

Seen on an individual eigenvector level,

$$\Sigma_\mu u = \lambda u \implies \Sigma_{T_{\#}\mu}(Ru) = R\Sigma_\mu R^\top (Ru) = R(\Sigma_\mu u) = \lambda(Ru),$$

Thus, the eigenvalues of $\Sigma_{T_{\#}\mu}$ are equal to those of $\Sigma_\mu$ and its eigenvectors are interpreted as orthogonally transformed versions of those of $\mu$. For the eigenvectors corresponding to the non-zero eigenvalues, the transformation is unique up to sign. The two covariance matrices have the same distribution of eigenvalues (unique non-zero eigenvalues, some number of zero eigenvalues), so the only ambiguity in finding a non-zero eigenvalue eigenvector is the sign. For the zero-eigenvalue eigenvectors, which may have multiplicity, there is more to say.

For the zero-eigenvalue eigenspace, any orthonormal basis spans the kernel. Projections of $\mu$ onto any direction in this subspace yield Dirac masses at zero. Although there is some ambiguity in choosing them, we only use these eigenvectors to induce distributions on the real line, so the end effect is the same. Also, the sign ambiguity doesn't matter either (reflection of a Dirac mass at zero is still a Dirac mass at 0).

For the non-zero eigenvalue eigenvectors, the projection of rotated data onto rotated eigenvectors induces the same distribution. That is,

for all $x \in \mathbb{R}^d:$    $\langle Rx,\ Ru \rangle = \langle x,\ u \rangle,$  so for any sample $\{x_i\},\ \{\langle Rx_i,\ Ru \rangle\}_i = \{\langle x_i,\ u \rangle\}_i$

This assumes that we chose the optimal relative sign difference, because otherwise one of these multisets is reflected across 0. The element in the cost matrix for this pairing removes the ambiguity regarding the sign and recovers the correct relative sign between them. That is, for projections onto non-zero eigenvalue eigenvectors, we knew the induced distributions were unique up to sign, and $s$ handles the relative difference in sign.

$$c(\pm u, \pm Ru) = \min_{s \in \{\pm 1\}} W_2^2 \left( \{\langle x_i, u \rangle\}_{i=1}^n,\ \{s\langle Rx_i, Ru \rangle\}_{i=1}^n \right)$$

Notationally, what we are illustrating is that there is sign ambiguity in how each axis is obtained from PCA (up to sign), but regardless of that, the cost matrix entry will be the same.

$W_2$ is a metric, so $W_2^2$ is 0 if and only if the two multisets are equal. Thus, for one of these two terms in the minimization, $W_2^2$ will be 0. This is because Wasserstein is invariant under simultaneous reflection, so we only need to consider two cases instead of four.

As stated earlier, the zero eigenvalues all yield Dirac masses at 0, and the cost matrix entry between them will be 0.

Thus, if $\pi(i)$ is defined to pair axes with the same eigenvalue to axes of the same eigenvalue, each $c_{i,\pi(i)}$ will be 0. This is feasible because they have the same eigenvalue distribution. This can be done uniquely for the top $r$ eigenvectors, and in any such way for the remaining indices $r + 1, ...d$. The end result is that identical (up to sign) multisets are paired together, and scored as 0 cost, and any Diracs are paired together for 0 cost.

$$c_{i,\pi(i)} = \min_{s \in \{\pm 1\}} W_2^2\Big(\{\langle x_j, u_i\rangle\}_j,\ \{s\langle Rx_j, v_{\pi(i)}\rangle\}_j\Big) = 0,$$

Thus, $\mathrm{D}^2(\mu, T_\#\mu) = 0 \implies \mathrm{D}(\mu, T_\#\mu) = 0$ as

$$\mathrm{D}^2 \le \frac{1}{k}\sum_{j=1}^{k} c\big(u_j,\ v_{\pi(j)}\big) = 0$$

as we constructed one such signed permutation that is minimized over and RISWIE is non-negative.

Note that we can take only the top $k$ eigenvectors (truncated SVD) and still obtain rigid-invariance by defining the same bijection $\pi$ but truncating the two sets of eigenvectors, keeping only the top $k$ by eigenvalue in each. This will also result in a RISWIE distance of 0.

We have directly shown the special case that when two distributions differ by a rigid transformation that their distance is 0. It is a simple generalization to show that arbitrary rigid transformations applied to one of two different distributions do not change the RISWIE distance.

That is, for two measures $\mu, \nu$ (still making simple non-zero covariance eigenvalue assumptions), any for any rigid maps $T(x) = Rx,\ S(y) = Qy,$

$$D(\mu, \nu) = D(T_\#\mu, \nu) = D(\mu, S_\#\nu) = D(T_\#\mu, S_\#\nu)$$

This is because the RISWIE distance is just a function of the 1D marginals. The 1D marginals are actually the same up to sign for the same distribution before and after a rigid transformation. Thus, when we do axis-pairing, it doesn't matter whether a distribution was rigidly transformed or not. RISWIE will optimize over signs and remove that ambiguity.

**Diffusion Maps:**   Define the kernel

$$K_{ij} = k\left(\frac{\|x_i - x_j\|^2}{\varepsilon}\right) \qquad (\text{e.g. } k(s) = e^{-s})$$

Rigid transformations preserve pairwise distances

$$\|T(x_i) - T(x_j)\| = \|Rx_i + t - (Rx_j + t)\| = \|R(x_i - x_j)\| = \|x_i - x_j\|$$

Consequently, the construction of the kernel matrix itself is rigid-invariant. If we called the kernel matrix $K'$ (build from $\{T(x_i)\}$), then $K' = K$.

As such, given that the entire diffusion procedure (writing the degree matrix $E$, Laplacian $L$, EVD, etc) is entirely derived from the kernel matrix, the embedded distributions should be exactly the same.

$$E' = \mathrm{diag}(K\mathbf{1}) = E, \qquad L'_{\mathrm{rw}} = E^{-1}K = L_{\mathrm{rw}}, \quad L'_{\mathrm{sym}} = I - E^{-1/2}KE^{-1/2} = L_{\mathrm{sym}}.$$

Let $L_{sym}\Phi = \Phi\Lambda$ be an be an eigendecomposition.

Point $i$ is embedded with diffusion coordinates

$$\Psi_t(i) = \left(\lambda_1^t\,\phi_1(i),\ldots,\lambda_k^t\,\phi_k(i)\right)^\top$$

for some fixed time $t$.

Given that the construction of $L_{sym}$ is rigid-invariant, the eigenvectors returned by an eigensolver for $L_{sym}$ and $L'_{sym}$ should be the same. Whether this is true in practice depends on the implementation of numerical eigensolvers. It would suffice to assume a simple spectrum, which would ensure that the eigenvectors are unique up to sign, but it is not necessary. As such, we only assume that the eigensolver used is deterministic.

Thus, following the same argument as for PCA, if the $k$ 1D distributions are the same whether or not a rigid transformation is applied to the distribution, then the RISWIE distance between any two shapes does not depend on arbitrary rigid transformations applied to them. So $D(\mu,\nu) = D(T_\#\mu, S_\#\nu)$ where diffusion map embeddings in $D$ are implicitly used as well.

$\square$

*Proof of Theorem 2.* Let $\mathcal{E}$ be any deterministic $k$-dimensional embedding procedure. Then for any $X, Y, Z \in \mathcal{P}_2(\mathbb{R}^d)$, the RISWIE distance satisfies:

  (i) Non-negativity: $D(X,Y) \geq 0$,

  (ii) Symmetry: $D(X,Y) = D(Y,X)$,

  (iii) Triangle inequality: $D(X,Z) \leq D(X,Y) + D(Y,Z)$,

The square root of the average of $W_2^2$ distances is non-negative and symmetric.

Let $R_{XY} = \mathrm{argmin}_{R\in\mathcal{O}_k^\pm} \dfrac{1}{k}\sum_{j=1}^{k} W_2^2\left(\alpha_j, \beta_{Rj}\right), \quad R_{YZ} = \mathrm{argmin}_{R\in\mathcal{O}_k^\pm} \dfrac{1}{k}\sum_{j=1}^{k} W_2^2\left(\beta_j, \gamma_{Rj}\right).$

Define the composite signed permutation $R_{XZ} = R_{YZ}\,R_{XY} \in \mathcal{O}_k^\pm$. For each $j$, let

$$u_j = W_2\left(\alpha_j, \beta_{R_{XY}j}\right), \quad v_j = W_2\left(\beta_{R_{XY}j}, \gamma_{R_{XZ}j}\right), \quad w_j = W_2\left(\alpha_j, \gamma_{R_{XZ}j}\right).$$

By the one-dimensional triangle inequality,

$$w_j = W_2\left(\alpha_j, \gamma_{R_{XZ}j}\right) \leq W_2\left(\alpha_j, \beta_{R_{XY}j}\right) + W_2\left(\beta_{R_{XY}j}, \gamma_{R_{XZ}j}\right) = u_j + v_j.$$

Hence componentwise $w \leq u + v$, so

$$\|w\|_2 \leq \|u+v\|_2 \leq \|u\|_2 + \|v\|_2,$$

and dividing by $\sqrt{k}$ gives

$$\sqrt{\frac{1}{k}\sum_{j=1}^{k} w_j^2} \leq \sqrt{\frac{1}{k}\sum_{j=1}^{k} u_j^2} + \sqrt{\frac{1}{k}\sum_{j=1}^{k} v_j^2}.$$

Since $R_{XZ}$ is only a candidate for the minimization defining $D(X,Z)$,

$$\mathrm{D}(X,Z) = \min_{R\in\mathcal{O}_k^\pm} \sqrt{\frac{1}{k}\sum_{j=1}^{k} W_2^2(\alpha_j, \gamma_{Rj})} \leq \sqrt{\frac{1}{k}\sum_{j=1}^{k} w_j^2} \leq \mathrm{D}(X,Y) + \mathrm{D}(Y,Z).$$

$\square$

*Proof of Theorem 3.* Without loss of generality, consider the centered versions $A \sim \mathcal{N}(0, \Sigma_A)$ and $B \sim \mathcal{N}(0, \Sigma_B)$, as RISWIE is translation-invariant.

Projecting $A \sim \mathcal{N}(0, \Sigma_A)$ onto its $i$th PCA axis $u_i$ yields a one-dimensional Gaussian, since $u_i^\top x \sim \mathcal{N}(0, \lambda_i^A)$. Similarly, projecting $B \sim \mathcal{N}(0, \Sigma_B)$ onto its $j$th PCA axis $v_j$ yields, with $v_j^\top y \sim \mathcal{N}(0, \lambda_j^B)$. Take

$$a_i := \sqrt{\lambda_i^A}$$

and $b_j := \sqrt{\lambda_j^B}$. It is known that the squared Wasserstein-2 distance between $\mathcal{N}(0, \lambda_i^A)$ and $\mathcal{N}(0, \lambda_j^B)$ is $(a_i - b_j)^2$.

Thus, the RISWIE cost for a permutation $\pi \in S_d$ is

$$C(\pi) := \frac{1}{d} \sum_{i=1}^{d} (a_i - b_{\pi(i)})^2.$$

We claim this is minimized when both vectors are sorted in increasing order (i.e., $\pi^* = \mathrm{id}$). Note that $a_1 \leq \cdots \leq a_d$ (the $a_i$ are sorted).

Indeed, consider swapping two positions, say $i < j$, and compare the change in costs between the two permutations:

$$
\begin{aligned}
\Delta &:= \left[ (a_i - b_j)^2 + (a_j - b_i)^2 \right] - \left[ (a_i - b_i)^2 + (a_j - b_j)^2 \right] \\
&= \left[ a_i^2 - 2a_i b_j + b_j^2 + a_j^2 - 2a_j b_i + b_i^2 \right] - \left[ a_i^2 - 2a_i b_i + b_i^2 + a_j^2 - 2a_j b_j + b_j^2 \right] \\
&= \left[ -2a_i b_j + b_j^2 - 2a_j b_i + b_i^2 \right] - \left[ -2a_i b_i + b_i^2 - 2a_j b_j + b_j^2 \right] \\
&= -2a_i b_j + b_j^2 - 2a_j b_i + b_i^2 + 2a_i b_i - b_i^2 + 2a_j b_j - b_j^2 \\
&= 2a_i(b_i - b_j) + 2a_j(b_j - b_i) \\
&= 2(a_j - a_i)(b_j - b_i).
\end{aligned}
$$

If $b_j < b_i$ (an inversion relative to the $a$−order, then $b_j - b_i < 0$ and hence $\Delta \leq 0$. So swapping $b_i$, $b_j$ for the increasing sorted order does not increase the cost, and strictly decreases it unless $a_i = a_j$.

Thus, given any permutation, it can be improved by swapping inverted adjacent pairs. The only time we can't improve a solution is there are no inversions, i.e. when

$$b_{\pi(1)} \leq b_{\pi(2)} \leq \cdots \leq b_{\pi(d)}$$

Since any permutation can be reduced to the identity via a sequence of such swaps, and each swap never increases the cost, the minimal cost is achieved by the identity permutation:

$$C(\mathrm{id}) = \frac{1}{d} \sum_{i=1}^{d} (a_i - b_i)^2.$$

Therefore,

$$\mathrm{D}_G^2(A, B) = \frac{1}{d} \|\mathbf{a} - \mathbf{b}\|_2^2,$$

as claimed. Here, we denote $D_G$ to be the Gaussian closed form. $\qquad\square$

*Proof of Theorem 4.* We use the bounds from (Salmona et al., 2022):

**Upper Bounding RISWIE:**

$$LGW_2^2(A, B) = 4(\mathrm{tr}(\Lambda_A) - \mathrm{tr}(\Lambda_B))^2 + 4(\|\Lambda_A\|_F - \|\Lambda_B\|_F)^2 + 4\|\Lambda_A - \Lambda_B\|_F^2,$$

$$GGW_2^2(A, B) = 4(\mathrm{tr}(\Lambda_A) - \mathrm{tr}(\Lambda_B))^2 + 8\|\Lambda_A - \Lambda_B\|_F^2 + 8(\|\Lambda_A\|_F^2 - \|\Lambda_A^{(n)}\|_F^2).$$

Here, $LGW$ and $GGW$ are lower and upper bounds for $GW_2^2$. The results from Salmona et al. (2022) are general and apply to Gaussian measures defined on Euclidean spaces of differing dimensions. For clarity and interpretability, however, we focus on the case where both distributions lie

in the same ambient space. As such, we have already dropped an additional term from the original formulation, which accounted for the difference in Frobenius norm between the full covariance eigenvalue matrix and its truncation to the lower-dimensional space. This term vanishes in our setting since both distributions lie in the same ambient space, and no truncation is required.

Let $a_i = \sqrt{\lambda_i^A}$, $b_i = \sqrt{\lambda_i^B}$, and $\alpha = \min_i(a_i + b_i)$. Note that $(\lambda_i^A - \lambda_i^B)^2 = (a_i + b_i)^2(a_i - b_i)^2 \geq \alpha^2(a_i - b_i)^2$ for all $i$.

Therefore,

$$\|\Lambda_A - \Lambda_B\|_F^2 = \sum_{i=1}^d (\lambda_i^A - \lambda_i^B)^2 \geq \alpha^2 \sum_{i=1}^d (a_i - b_i)^2 = d\alpha^2 D_G^2(A, B).$$

Since all other terms in $LGW_2^2$ are nonnegative,

$$LGW_2^2(A, B) \geq 4\|\Lambda_A - \Lambda_B\|_F^2 \geq 4d\alpha^2 D_G^2(A, B).$$

Similarly,

$$GGW_2^2(A, B) \geq 8\|\Lambda_A - \Lambda_B\|_F^2 \geq 8d\alpha^2 D_G^2(A, B).$$

Hence,

$$D_G^2(A, B) \leq \frac{GGW_2^2(A, B)}{8d\alpha^2}.$$

Additionally, Salmona et al. (2022) shows a bound on the difference between the upper and lower bounds:

$$GGW_2^2(A, B) - LGW_2^2(A, B) \leq 8\|\Sigma_A\|_F\|\Sigma_B\|_F \left(1 - \frac{1}{\sqrt{d}}\right).$$

Because $GW_2^2(A, B) \leq GGW_2^2(A, B)$, and $LGW_2^2(A, B) \leq GW_2^2(A, B)$, we may write

$$GGW_2^2(A, B) = GW_2^2(A, B) + (GGW_2^2(A, B) - GW_2^2(A, B))$$
$$\leq GW_2^2(A, B) + (GGW_2^2(A, B) - LGW_2^2(A, B)).$$

Plugging this into the previous bound,

$$D_G^2(A, B) \leq \frac{GW_2^2(A, B)}{8d\alpha^2} + \frac{GGW_2^2(A, B) - LGW_2^2(A, B)}{8d\alpha^2}$$
$$\leq \frac{GW_2^2(A, B)}{8d\alpha^2} + \frac{\|\Sigma_A\|_F\|\Sigma_B\|_F}{d\alpha^2} \left(1 - \frac{1}{\sqrt{d}}\right).$$

For the second bound, note that for all $i$,

$$(a_i - b_i)^2 = \left(\sqrt{\lambda_i^A} - \sqrt{\lambda_i^B}\right)^2 \leq \left|\lambda_i^A - \lambda_i^B\right|,$$

since by the factorization $a_i^2 - b_i^2 = (a_i - b_i)(a_i + b_i)$ and the triangle inequality,

$$|a_i - b_i| \leq |a_i + b_i| \implies (a_i - b_i)^2 \leq |a_i^2 - b_i^2| = |\lambda_i^A - \lambda_i^B|.$$

We note that $|a_i - b_i| \leq |a_i + b_i|$ holds specifically because we are dealing with non-negative $a_i$ and $b_i$.

Thus,

$$D_G^2(A, B) = \frac{1}{d}\sum_{i=1}^d (a_i - b_i)^2 \leq \frac{1}{d}\sum_{i=1}^d |\lambda_i^A - \lambda_i^B|.$$

By Cauchy–Schwarz,

$$\sum_{i=1}^d |\lambda_i^A - \lambda_i^B| \leq \sqrt{d}\left(\sum_{i=1}^d (\lambda_i^A - \lambda_i^B)^2\right)^{1/2} = \sqrt{d}\|\Lambda_A - \Lambda_B\|_F.$$

Thus,

$$\mathrm{D}_G^2(A, B) \leq \frac{1}{\sqrt{d}} \|\Lambda_A - \Lambda_B\|_F.$$

But $GW_2^2(A, B) \geq 4\|\Lambda_A - \Lambda_B\|_F^2 + 4(\mathrm{tr}(\Lambda_A) - \mathrm{tr}(\Lambda_B))^2 + 4(\|\Lambda_A\|_F - \|\Lambda_B\|_F)^2$, so

$$\|\Lambda_A - \Lambda_B\|_F^2 \leq \frac{1}{4}\left( GW_2^2(A, B) - 4(\mathrm{tr}(\Lambda_A) - \mathrm{tr}(\Lambda_B))^2 - 4(\|\Lambda_A\|_F - \|\Lambda_B\|_F)^2 \right).$$

Therefore,

$$\|\Lambda_A - \Lambda_B\|_F \leq \frac{1}{2}\sqrt{GW_2^2(A, B) - 4(\mathrm{tr}(\Lambda_A) - \mathrm{tr}(\Lambda_B))^2 - 4(\|\Lambda_A\|_F - \|\Lambda_B\|_F)^2}.$$

Putting this together,

$$\mathrm{D}_G^2(A, B) \leq \frac{1}{2\sqrt{d}}\sqrt{GW_2^2(A, B) - 4\left(\mathrm{tr}(\Lambda_A) - \mathrm{tr}(\Lambda_B)\right)^2 - 4\left(\|\Lambda_A\|_F - \|\Lambda_B\|_F\right)^2}.$$

**Lower Bounding RISWIE:**   We define

$$\beta := \max_{1 \leq i \leq d}(a_i + b_i).$$

Again from Salmona et al. (2022), as we work in the case where the distributions are in the same dimensional space:

$$GGW_2^2(A, B) = 4\left(\mathrm{tr}(\Lambda_A) - \mathrm{tr}(\Lambda_B)\right)^2 + 8\|\Lambda_A - \Lambda_B\|_F^2$$

For each $i$,

$$\lambda_i^A - \lambda_i^B = a_i^2 - b_i^2 = (a_i - b_i)(a_i + b_i),$$

so

$$|a_i - b_i| = \frac{|\lambda_i^A - \lambda_i^B|}{a_i + b_i} \geq \frac{|\lambda_i^A - \lambda_i^B|}{\beta},$$

because $a_i + b_i \leq \beta$ by definition of $\beta$. Squaring and summing,

$$\sum_{i=1}^d (a_i - b_i)^2 \geq \frac{1}{\beta^2}\sum_{i=1}^d (\lambda_i^A - \lambda_i^B)^2 = \frac{1}{\beta^2}\|\Lambda_A - \Lambda_B\|_F^2.$$

Hence, by Theorem 3,

$$D_G^2(A, B) = \frac{1}{d}\sum_{i=1}^d (a_i - b_i)^2 \geq \frac{1}{d\,\beta^2}\|\Lambda_A - \Lambda_B\|_F^2. \tag{$*$}$$

From the Gaussian upper bound,

$$GW_2^2(A, B) \leq GGW_2^2(A, B) = 4\left(\mathrm{tr}(\Lambda_A) - \mathrm{tr}(\Lambda_B)\right)^2 + 8\|\Lambda_A - \Lambda_B\|_F^2.$$

By Cauchy–Schwarz,

$$\left(\mathrm{tr}(\Lambda_A) - \mathrm{tr}(\Lambda_B)\right)^2 = \left(\sum_{i=1}^d (\lambda_i^A - \lambda_i^B)\right)^2 \leq d\sum_{i=1}^d (\lambda_i^A - \lambda_i^B)^2 = d\,\|\Lambda_A - \Lambda_B\|_F^2.$$

Therefore,

$$GGW_2^2(A, B) \leq 4d\,\|\Lambda_A - \Lambda_B\|_F^2 + 8\,\|\Lambda_A - \Lambda_B\|_F^2 = 4(d+2)\,\|\Lambda_A - \Lambda_B\|_F^2,$$

and hence

$$GW_2^2(A, B) \leq 4(d+2)\,\|\Lambda_A - \Lambda_B\|_F^2. \tag{$**$}$$

From $(*)$ we have

$$\|\Lambda_A - \Lambda_B\|_F^2 \leq d\,\beta^2\,D_G^2(A, B).$$

Substituting into $(**)$ gives

$$GW_2^2(A, B) \leq 4(d+2)\, d\, \beta^2\, D_G^2(A, B).$$

Rearranging,

$$D_G^2(A, B) \geq \frac{GW_2^2(A, B)}{4\, d(d+2)\, \beta^2},$$

and taking square roots yields

$$D_G(A, B) \geq \frac{GW_2(A, B)}{2\, \beta\, \sqrt{d(d+2)}}.$$

$\square$

**Corollary 1** (Identity of Indiscernibles for Gaussians). *Under the same setting as above, $D_G(A, B) = 0$ if and only if there exists an orthogonal matrix $R$ and translation $t$ such that $B$ is the distribution of $RX + t$ for $X \sim A$.*

*Proof.* From Theorem 3,

$$D_G^2(A, B) = \frac{1}{d} \sum_{j=1}^{d} \left( \sqrt{\lambda_j^A} - \sqrt{\lambda_j^B} \right)^2$$

after labeling the eigenvalues in sorted order. Hence,

$$D_G(A, B) = 0$$

if and only if the eigenvalues of $\Sigma_A$ and $\Sigma_B$ agree up to permutation, i.e., there exists a permutation $\pi$ such that

$$\lambda_j^A = \lambda_{\pi(j)}^B, \qquad j = 1, \ldots, d.$$

Assume, without loss of generality, that $A$ and $B$ are centered (otherwise, subtract their means). Write

$$\Sigma_A = U_A \Lambda U_A^\top, \qquad \Sigma_B = U_B \Lambda U_B^\top,$$

where $\Lambda$ is the common diagonal matrix of eigenvalues (after reordering). Then

$$\Sigma_B = U_B \Lambda U_B^\top = U_B U_A^\top \Sigma_A U_A U_B^\top.$$

Setting

$$T := U_B U_A^\top,$$

we see that $T$ is orthogonal and

$$\Sigma_B = T \Sigma_A T^\top.$$

Therefore, if $X \sim A$, then $TX$ is a centered Gaussian with covariance $\Sigma_B$, so $TX \sim B$. Thus, $B$ is the law of $TX$ for some orthogonal $T$.

Conversely, if $B$ is the distribution of $TX + t$ for some orthogonal $T$ and $X \sim A$, then

$$\Sigma_B = T \Sigma_A T^\top.$$

Orthogonal conjugation (a similarity transform) preserves the multiset of eigenvalues, so $\Sigma_A$ and $\Sigma_B$ have the same eigenvalues. By Theorem 3, this implies

$$D_G(A, B) = 0.$$

$\square$

**Theorem 5** (Stability of RISWIE under Gaussian Covariance Perturbations). *If $\Sigma' = \Sigma_X + E$ with $E = E^\top$ and all eigenvalues of $\Sigma_X, \Sigma'$ are $\geq \lambda_{\min} > 0$, then*

$$D_G(X, X') \leq \frac{\|E\|_2}{2\sqrt{\lambda_{\min}}}.$$

*Proof.* By Weyl's theorem for symmetric matrices (as discussed by Shamrai (2025)), for each $i = 1, \ldots, d$,

$$|\lambda_i(\Sigma') - \lambda_i(\Sigma_X)| \leq \|\Sigma' - \Sigma_X\|_2 = \|E\|_2 \leq \eta,$$

where we set $\eta := \|E\|_2$.

Consider the function $f(x) = \sqrt{x}$ for $x \geq 0$. By the mean value theorem, for each $i$, there exists $\xi_i$ between $\lambda_i(\Sigma_X)$ and $\lambda_i(\Sigma')$ such that

$$\left| \sqrt{\lambda_i(\Sigma')} - \sqrt{\lambda_i(\Sigma_X)} \right| = f'(\xi_i) \cdot |\lambda_i(\Sigma') - \lambda_i(\Sigma_X)|.$$

Since $f'(x) = \frac{1}{2\sqrt{x}}$ and all eigenvalues of $\Sigma_X$ and $\Sigma'$ are at least $\lambda_{\min}$, we have $\xi_i \geq \lambda_{\min}$, and $f'(\xi_i)$ is decreasing, so

$$f'(\xi_i) = \frac{1}{2\sqrt{\xi_i}} \leq \frac{1}{2\sqrt{\lambda_{\min}}}.$$

Therefore,

$$\left| \sqrt{\lambda_i(\Sigma')} - \sqrt{\lambda_i(\Sigma_X)} \right| \leq \frac{1}{2\sqrt{\lambda_{\min}}} \cdot \eta.$$

Let $\sigma_i := \sqrt{\lambda_i(\Sigma_X)}$, $\sigma'_i := \sqrt{\lambda_i(\Sigma')}$, and collect them as vectors $\sigma = (\sigma_1, \ldots, \sigma_d)$, $\sigma' = (\sigma'_1, \ldots, \sigma'_d)$.

Then,

$$\|\sigma' - \sigma\|_2 \leq \sqrt{\sum_{i=1}^{d} \left( \frac{\eta}{2\sqrt{\lambda_{\min}}} \right)^2} = \frac{\eta}{2\sqrt{\lambda_{\min}}} \sqrt{d},$$

so

$$D_G(X, X') \leq \frac{1}{\sqrt{d}} \cdot \frac{\eta}{2\sqrt{\lambda_{\min}}} \sqrt{d} = \frac{\eta}{2\sqrt{\lambda_{\min}}}.$$

More generally, if the lower bound for each eigenvalue is $\min(\lambda_i(\Sigma_X), \lambda_i(\Sigma'))$, then by the same reasoning,

$$D_G(X, X') \leq \frac{\eta}{2\sqrt{d}} \sqrt{\sum_{i=1}^{d} \frac{1}{\min(\lambda_i(\Sigma_X), \lambda_i(\Sigma'))}}.$$

$\square$

**Theorem 6** (Consistency of empirical RISWIE). *Let $\hat{\mu}_n, \hat{\nu}_n$ denote empirical measures of size $n$ drawn i.i.d. from $\mu, \nu \in \mathcal{P}_2(\mathbb{R}^d)$, respectively. For each $n$, let $\{\phi_{n,j}\}_{j=1}^{k}$ and $\{\psi_{n,j}\}_{j=1}^{k}$ be (possibly data-dependent) embedding functions used by RISWIE, and let $\{\phi_j\}_{j=1}^{k}$, $\{\psi_j\}_{j=1}^{k}$ be limiting embedding functions.*

*Assume that for each $j = 1, \ldots, k$: (i) the pushforward measures converge almost surely,*

$$(\phi_{n,j})_{\#}\hat{\mu}_n \Rightarrow (\phi_j)_{\#}\mu \quad and \quad (\psi_{n,j})_{\#}\hat{\nu}_n \Rightarrow (\psi_j)_{\#}\nu,$$

*and (ii) their second moments converge almost surely.*

*Then*

$$D(\hat{\mu}_n, \hat{\nu}_n) \xrightarrow{a.s.} D(\mu, \nu) \quad as \ n \to \infty,$$

*where $D(\mu, \nu)$ is defined using the limiting embeddings $\{\phi_j\}, \{\psi_j\}$.*

*Proof.* Fix $R \in \mathcal{O}_k^{\pm}$ and $j \in \{1, \ldots, k\}$. By the embedding-consistency assumption, the one-dimensional pushforwards $(\phi_{n,j})_{\#}\hat{\mu}_n$ and $(\psi_{n,Rj})_{\#}\hat{\nu}_n$ converge weakly almost surely to $(\phi_j)_{\#}\mu$ and $(\psi_{Rj})_{\#}\nu$, respectively, and their second moments converge almost surely. Since $W_2$ on $\mathbb{R}$ is continuous under weak convergence plus convergence of second moments, it follows that

$$W_2((\phi_{n,j})_{\#}\hat{\mu}_n, \ (\psi_{n,Rj})_{\#}\hat{\nu}_n) \xrightarrow{a.s.} W_2((\phi_j)_{\#}\mu, \ (\psi_{Rj})_{\#}\nu).$$

Averaging over $j = 1, \dots, k$ preserves almost sure convergence. Finally, since $\mathcal{O}_k^{\pm}$ is finite, taking the minimum over $R \in \mathcal{O}_k^{\pm}$ also preserves almost sure convergence. Therefore,

$$D(\hat{\mu}_n, \hat{\nu}_n) \xrightarrow{\text{a.s.}} D(\mu, \nu),$$

as claimed. □

*Remark* 1 (Bias of the empirical RISWIE estimator). Let $\mu$ be Borel probability measure with finite second moments. Then, $D(\mu, \mu) = 0$, but

$$\mathbb{E}\big[D(\hat{\mu}_n, \hat{\mu}'_n)\big] > 0,$$

where $\hat{\mu}'_n$ is another independent sample of $\mu$.

*Proof.* We have $D(\mu, \mu) = 0$, since projecting and optimally matching each direction trivially yields zero cost. However, the independent empirical marginals $\hat{\alpha}_j$ and $\hat{\alpha}'_j$ almost surely differ, and thus $W_2^2(\hat{\alpha}_j, \hat{\alpha}'_j) > 0$ almost surely for each $j$. Therefore, averaging and minimizing still yields strictly positive expectation:

$$\mathbb{E}\big[D(\hat{\mu}_n, \hat{\mu}'_n)\big] > 0.$$

□

