# OpenReview forum: "Rigid Invariant Sliced Wasserstein via Independent Embeddings"
_ICLR.cc/2026/Workshop/GRaM — ICLR 2026 Workshop GRaM Poster_

### Official Review · Reviewer_hKbM · 2026-02-08
**A Promising, Scalable Rigid-Invariant Distance, but Differentiability Needs More Spotlight**

**Rating:** 7
**Confidence:** 4

**Review:**

The paper introduces Rigid-Invariant Sliced Wasserstein via Independent Embeddings (RISWIE), a novel distance metric designed to compare probability measures modulo rigid transformations. By leveraging data-dependent embeddings (like PCA or Diffusion Maps) and aligning 1D projected distributions via optimal signed permutations, RISWIE achieves nearly linear computational complexity. This is a significant improvement over classical rigid-invariant distances like Gromov-Wasserstein (GW) or Procrustes-Wasserstein (PW), which are typically NP-hard or cubic in complexity. The authors provide theoretical bounds relating RISWIE to GW and demonstrate superior runtime and competitive accuracy on 3D mesh (FAUST) and cellular imaging (HuBMAP) datasets.Strengths:Exceptional Computational Efficiency: The primary contribution is the drastic reduction in computational cost. Moving from the $O(n^3)$ complexity of approximate GW solvers to the nearly linear time of RISWIE (using PCA embeddings) is a major breakthrough. This scalability makes rigid-invariant transport feasible for large-scale geometric data analysis.Strong Empirical Performance: The experiments on the FAUST and HuBMAP datasets convincingly demonstrate that the efficiency gain does not come at the cost of accuracy. In clustering and alignment tasks, RISWIE matches or outperforms significantly more expensive baselines.Theoretical Grounding: The paper provides solid theoretical properties, including the proof that RISWIE is a pseudometric, its rigid-invariance guarantees, and interesting closed-form bounds relating it to the Gromov-Wasserstein distance for Gaussian measures.Potential Impact: As noted in the motivation, finding a metric that is both fast and rigid-invariant is a "holy grail" for many geometric learning tasks.Weaknesses & Critique:Differentiability and Applicability as a Loss Function:The standard RISWIE formulation relies on sorting (for 1D Wasserstein) and the Hungarian algorithm (for axis alignment), which are non-differentiable operations (or have zero gradients almost everywhere).While the authors address this in Section 5 and Appendix A.6 by introducing "SRISWIE" (Soft RISWIE using entropic regularization), this crucial feature is relegated to the appendix.For this metric to achieve widespread adoption in representation learning (e.g., as a contrastive loss or reconstruction loss in deep learning, potentially rivaling MSE or Cross-Entropy in geometric contexts), fully differentiable implementation is essential. The current main text focuses heavily on the discrete, hard-assignment version.Dependence on Embedding Stability: The method's reliability hinges on the stability of the embedding axes (e.g., PCA eigenvectors). While the authors discuss handling sign ambiguities and permutations, the paper could better address cases with small eigengaps where axes might mix, potentially causing instability in the gradients even in the soft version.Questions for Authors:Gradient Quality of SRISWIE: In Appendix A.6, you propose SRISWIE for differentiable optimization. Can you provide any preliminary results or intuition on the stability of the gradients provided by SRISWIE compared to standard OT losses? Does the "soft" alignment vanish or explode during training?Performance Trade-off: How does the Soft-RISWIE (SRISWIE) perform on the benchmark tasks (FAUST/HuBMAP) compared to the "Hard" RISWIE? If the performance is comparable, it would strengthen the case for using this as a general-purpose loss function.Sensitivity to Outliers: Since PCA is sensitive to outliers, does the reliance on PCA embeddings make RISWIE less robust to noise compared to GW (which uses internal relational structures)?Reason for Recommendation:I recommend a Weak Accept. The paper proposes a highly valuable tool for geometric data analysis that solves the efficiency bottleneck of rigid-invariant optimal transport. The idea is novel, mathematically grounded, and empirically effective.However, I stopped short of a "Strong Accept" because the potential for this metric to be used as a differentiable loss function—which is arguably its most exciting application for the deep learning community—is treated as an afterthought in the Appendix. If the authors can demonstrate that the differentiable version (SRISWIE) is robust and effective, this work would have significant impact beyond just a static distance measure.

**Pmlr Suitability:**

Yes

---

### Official Review · Reviewer_LxQA · 2026-02-20
**Meaningful theoretical contribution**

**Rating:** 7
**Confidence:** 3

**Review:**

The paper tackles the problem of comparing two distributions when they may be subject to rigid motions. Standard OT distances like Wasserstein and Sliced Wasserstein are not rigid-invariant, while rigid-invariant alternatives such as Gromov–Wasserstein (GW) are too computationally expensive. They propose a new pseudo-metric: Rigid-Invariant Sliced Wasserstein via Independent Embeddings (RISWIE): instead of aligning points or solving assignment over pairwise distances, they:

1. Embed each measure independently into a data-driven coordinate system (such as PCA coordinates).

2. Compare 1D pushforwards along embedding axes using the closed-form 1D Wasserstein.

3. Match axes across the two embeddings by minimizing the sum of 1D Wasserstein costs across signed permutations with Hungarian algorithm.

Strengths:
The paper is well written, with a remarkable theoretical analysis of the proposed pseudo-metric
Experiments show a clear superiority both in terms of accuracy and computation time

Major comments/remarks/questions:
Theorem 2 is limited to strictly deterministic embedding procedure, so PCA, Diffusion maps works fine but careful with stochastic embedding procedures in generative modeling for instance. This should be discussed

Minor comments/remarks:
For Theorem 4, alpha should be strictly positive
For corollary 1 in the supplementary material, “eigenvalues match up (up to permutation)” but it seems, R is orthogonal yes, but not necessarily a signed permutation
In the proof of Theorem 5, last line, if all eigenvalues are equal to lambda_min, then with the sum, D_G(X,X’) < \sqrt{d} \eta / 2\lambda_min which contradicts the upper bound just above

Recommendation:
Accept: really good theoretical contribution

**Pmlr Suitability:**

Yes

---

### Official Review · Reviewer_umei · 2026-02-23
**A computationally efficient rigid-invariant transport distance**

**Rating:** 7
**Confidence:** 3

**Review:**

**Summary**: This paper introduces RISWIE, a rigid-invariant transport distance that compares probability measures modulo rotations and reflections by combining data-dependent embeddings with signed-permutation alignment of 1D Wasserstein projections. The primary contribution is achieving near-linear computational complexity while retaining desirable invariance properties and meaningful connections to Gromov–Wasserstein (GW). The paper provides theoretical guarantees (pseudometric property, rigid-invariance under embedding conditions, Gaussian closed-form and bounds) and demonstrates strong empirical performance on FAUST and HuBMAP datasets.

**Strengths**:
* The computational improvement over GW and PW is substantial and practically significant.
* The theoretical development is thorough and well structured, particularly the Gaussian analysis and comparison bounds to GW. I did not notice any obvious mathematical inconsistencies
* Empirical results clearly demonstrate strong accuracy-runtime tradeoffs.
* The paper is generally well written and clearly motivated.

**Weaknesses**:
* The rigid-invariance guarantee depends on assumptions about embedding stability (e.g., distinct eigenvalues), and the role of the embedding in the theoretical results could be articulated more explicitly.
* How restrictive of an approximation is given by assuming full orthogonal alignment for the signed-permutation alignment strategy?
* The primary formulation is non-differentiable, while the differentiable relaxation appears in the appendix.

**Overall assessment**: The paper is a strong and technically solid contribution that meaningfully advances scalable rigid-invariant transport. The method is promising and well supported by both theory and experiments, though additional clarification of embedding assumptions and approximation effects would further strengthen the work.

**Pmlr Suitability:**

Yes

---

### Meta-Review · Area_Chair_uwRN · 2026-02-27

**Decision:**

Accept

**Metareview:**

Reviewers were unanimous in support of this paper proposing a distributional pseudometric with invariance properties and efficiency computational complexity.

**Relevance To Proceedings:**

Yes — suitable for PMLR (long paper)

**Relevance To Workshop:**

Yes — suitable for GRaM

---

### Decision · Program_Chairs · 2026-03-02

Accept (Poster)